# A Precise Characterization of SGD Stability Using Loss Surface Geometry

**Gregory Dexter**[1]**, Borja Ocejo**[2]**, Sathiya Keerthi**[2]**, Aman Gupta**[2]**,**
**Ayan Acharya**[2] **& Rajiv Khanna**[1] *
[1] Purdue University
[2] LinkedIn Corporation

## Abstract

Stochastic Gradient Descent (SGD) stands as a cornerstone optimization algorithm with proven real-world empirical successes but relatively limited theoretical understanding. Recent research has illuminated a key factor contributing to its practical efficacy: the implicit regularization it instigates. Several studies have investigated the *linear stability* property of SGD in the vicinity of a stationary point as a predictive proxy for sharpness and generalization error in overparameterized neural networks (Wu et al., 2022; Jastrzebski et al., 2019; Cohen et al., 2021). In this paper, we delve deeper into the relationship between linear stability and sharpness. More specifically, we meticulously delineate the necessary and sufficient conditions for linear stability, contingent on hyperparameters of SGD and the sharpness at the optimum. Towards this end, we introduce a novel *coherence measure* of the loss Hessian that encapsulates pertinent geometric properties of the loss function that are relevant to the linear stability of SGD. It enables us to provide a simplified sufficient condition for identifying linear instability at an optimum. Notably, compared to previous works, our analysis relies on significantly milder assumptions and is applicable for a broader class of loss functions than known before, encompassing not only mean-squared error but also cross-entropy loss.

## 1 Introduction

Stochastic Gradient Descent (SGD) is a fundamental optimization algorithm widely used in practice. In addition to its computational efficiency, there is irrefutable evidence of its superior generalization performance even on non-convex functions, including neural networks (Bottou, 1991). For large over-parameterized neural networks, the number of points to fit is often much less than the number of free parameters in the model. In this case, there is often a high-dimensional manifold of model weights that can perfectly fit the data (Cooper, 2021); hence, focusing solely on the ability of an optimizer to minimize the loss function ignores a central part of training such networks. The primary goal in a model is not to achieve high performance on a training data set but rather to achieve strong generalization performance on previously unseen data. Although we currently lack a comprehensive theoretical explanation for the empirical success of SGD in these models, a promising hypothesis suggests that SGD naturally applies a form of *implicit* regularization (Zhang et al., 2017; Neyshabur et al., 2015) when multiple optima are present (Keskar et al., 2017; Liu et al., 2020). This phenomenon guides the iterative process towards more favorable optima purely through algorithmic choices.

In order to measure this distinguishing favorability between more and less desirable optima, prior work has proposed the concept of *sharpness* at a minimum as an indicator of the generalization performance of the trained model. Lower sharpness is often indicative of better generalization performance (Hochreiter & Schmidhuber, 1997). There is a wealth of empirical work exploring the relationship between sharpness and generalization performance, particularly in networks trained with SGD, e.g., (Jiang et al., 2019; Jastrzebski et al., 2019; Andriushchenko et al., 2023; Wu et al., 2017; Chaudhari et al., 2017; Izmailov et al., 2018). Furthermore, these ideas have led to new optimizers which deliberately reduce sharpness and are observed to attain improved empirical performance (Behdin et al., 2023; Foret et al., 2020). Although the connection between sharpness

---

*Correspondence to Rajiv Khanna {*rajivak@purdue.edu*}

and generalization performance isn't precise or completely understood, the partial achievements of this theory has inspired several works, including ours, to investigate how SGD implicitly tends to converge to flatter optima.

Sharpness has been defined in several ways in prior literature, but most commonly, the sharpness of a trained neural network at a minimum is the maximum eigenvalue of the Hessian of the loss with respect to weights. Intuitively, one can see that if $\mathbf{w}^*$ is a stationary point of a smooth function $f(\mathbf{w})$ with Hessian $\mathbf{H}(\mathbf{w})$, then for perturbation $\mathbf{v}$ such that $\|\mathbf{v}\|_2 = \epsilon$, $f(\mathbf{w}^*+\mathbf{v}) < f(\mathbf{w}^*)+O(\epsilon^2)\cdot\lambda_1(\mathbf{H}(\mathbf{w}^*))$, where $\lambda_1(\cdot)$ denotes the maximum eigenvalue. This relation follows from the Taylor expansion of $f(\cdot)$ around $\mathbf{w}^*$, and we see that the sharpness at $\mathbf{w}^*$, i.e., $\lambda_1(\mathbf{H}(\mathbf{w}^*))$, determines how rapidly small perturbations to the weights $\mathbf{w}$ can increase the value of $f(\mathbf{w})$. In other words, model sharpness measures how robust the loss of the trained model is to small perturbations of the model parameters.

In this paper, our focus is on providing a precise characterization of how the SGD hyperparameters and properties of the loss function affect its implicit regularization of model sharpness. Towards this goal, we consider the *linearized dynamics* of SGD (defined in Section 2) close to the optimum. When $\mathbf{w}$ is close to $\mathbf{w}^\star$, it allows us to make a useful simplification by focusing on the quadratic approximation of the loss function. In particular, we consider mean-squared stability, that is, $\mathbf{w}^*$ is considered unstable if iterates of SGD diverge from $\mathbf{w}^*$ under the $\ell_2$-norm in expectation. Unlike differential equation-based approaches, which liken the SGD dynamics to a continuous flow for eliciting implicit regularization properties (Li et al., 2017; Xie et al., 2021), the linear stability analysis does not break down in the regime of large step sizes. Moreover, the linearized dynamics in gradient descent (GD) has been empirically validated to predict the sharpness of overparameterized neural networks due to the *Edge-of-Stability* phenomenon (Wu et al., 2018; Cohen et al., 2021). This behavior has also been observed in other optimizers (Cohen et al., 2022; Jastrzebski et al., 2019; Bartlett et al., 2022; Wen et al., 2022; Ujváry et al., 2022), lending further weight to this theoretical framework (Agarwala & Dauphin, 2023). While prior work has already considered the linear stability of SGD (Wu et al., 2022; 2018; Ma & Ying, 2021; Ziyin et al., 2023; Agarwala & Dauphin, 2023), our analysis provides substantial advancement over these prior results as we detail below.

**Contributions:**

• We offer an interpretable yet rigorously established sufficient condition to determine the instability of a point $\mathbf{w}^*$ under the linearized dynamics of SGD (Theorem 1). Importantly, unlike previous works, our bound applies to any additively decomposable loss function.

• Our sufficient condition hinges on a coherence measure $\sigma$, which we introduce to capture the relevant geometric characteristics of the loss surface around a minimum. This measure is intuitively connected to the Gram matrix of point-wise loss Hessians. We provide additional context and rationale for introducing this measure in Section 3.1.

• We demonstrate that our bound is nearly optimal across a natural range of SGD hyperparameters (Theorem 2). This implies that our analysis, which offers a sufficient condition for the stability of linearized SGD dynamics, is precise and closely aligned with the behavior of SGD across various choices for the coherence measure $\sigma$, batch size, and learning rate.

• In the course of deriving Theorem 1, we present an independently useful technical lemma (Lemma 4.1). This lemma provides sufficient conditions for (i) the divergence of linearized SGD dynamics and (ii) the convergence of linearized SGD dynamics toward the specified minima. An intriguing aspect of this lemma is that it suggests a multiplicative decoupling effect between the impact of batch size and learning rate on instability and the instability introduced by the geometry of the Hessian.

• Finally, we corroborate the validity of our theoretical findings through a series of experiments conducted on additively decomposable quadratic loss functions. Our experimental results align with our theory and underscore the significance of the Hessian coherence measure $\sigma$ in determining the conditions under which SGD dynamics diverge.

**Related Work:** While extensive research investigates the intricate relationship between optimization methods, generalization error, and sharpness, the prior work most relevant to ours focuses on a linear stability analysis of SGD. In this section, we briefly compare our results to related research. However, we defer a detailed comparison of our results until Section 3.2.1, which follows the formal introduction of the problem setup and the presentation of our primary theorem.

An important line of work in this area is that of Wu et al. (2018; 2022); Wu & Su (2023), which progressively distill more theoretical insight into how the linear dynamics of SGD affect final sharpness of a neural network. Our work goes beyond this in multiple ways. For sake of comparison, the result in this line of work most related to our contribution is Theorem 3.3 in Wu et al. (2022), and we restate this result in Appendix B. Our results are significantly more general than this theorem, in that our results apply to any general additively decomposable loss function, which answers the question raised in Agarwala & Dauphin (2023) on the behavior of SGD for cross-entropy loss.[1] Additionally, we guarantee a stronger form of divergence. Even with relaxed conditions and stronger implications, the condition of our theorem is practically easier to satisfy than Theorem 3.3 in Wu et al. (2022).

Other research delves into related questions, although our study may not align directly with them. For example, Jastrzebski et al. (2019) combine previous analysis by Wu et al. (2018) with additional assumptions about how the optimizer behaves. The paper demonstrates the impact of learning rate and batch size on sharpness throughout the training trajectories of SGD. Agarwala & Dauphin (2023) examine how batch size affects sharpness within SGD training trajectories, particularly in the context of second-order regression models. Ma & Ying (2021) provide a meticulous characterization of linear stability. However, this characterization might not be immediately interpretable and is primarily used to draw connections between behaviours of SGD and Sobolev regularization. Ziyin et al. (2023) focuses on the convergence and divergence of linearized dynamics in probability rather than in expected distance from the optimum, as considered by the other work we have mentioned.

## 2 PROBLEM FORMULATION

We consider the case where SGD is used to minimize an additively decomposable loss function $L(\mathbf{w}) = \frac{1}{n} \sum_{i=1}^{n} \ell_i(\mathbf{w})$, where each $\ell_i(\mathbf{w})$ is twice-differentiable and $\mathbf{w} \in \mathbb{R}^d$. Given learning rate $\eta > 0$ and batch size $B \in [n]$, the dynamics of SGD are defined by the recurrence $\mathbf{w}_{t+1} = \mathbf{w}_t - \frac{\eta}{B} \sum_{i \in \mathcal{S}} \nabla \ell_i(\mathbf{w}_t)$, where $\mathcal{S}$ is uniformly sampled from all $B$ sized subsets of $[n]$. To facilitate our probabilistic analysis, we apply two standard simplifications. First, we consider Bernoulli sampling rather than sampling with replacement so that $i \in \mathcal{S}$ with probability $B/n$ and the event $i \in \mathcal{S}$ is independent of the event $j \in \mathcal{S}$ for all $i \neq j$. Second, we consider the quadratic approximation to the loss around a fixed point $\mathbf{w}^*$, so that $\ell_i(\mathbf{w}) \approx \ell_i(\mathbf{w}^*) + (\mathbf{w} - \mathbf{w}^*)^T \nabla \ell_i(\mathbf{w}^*) + \frac{1}{2}(\mathbf{w} - \mathbf{w}^*)^T \nabla^2 \ell_i(\mathbf{w}^*)(\mathbf{w} - \mathbf{w}^*)$ (Wu et al., 2022; Ma & Ying, 2021). Since the dynamics of SGD are shift-invariant, we can assume $\mathbf{w}^* = \mathbf{0}$ without loss of generality. We restrict our attention to the case where $\mathbf{w}^*$ is a local minimum of $\ell_i(\cdot)$ for all $i \in [n]$. This assumption is particularly relevant in the context of overparameterized neural networks, where it is common for data to fit the model almost perfectly (Allen-Zhu et al., 2019).

In the described linearized setting, $\nabla \ell_i(\mathbf{w}) = \nabla_{\mathbf{w}}(\ell_i(\mathbf{w}^*) + \frac{1}{2} \mathbf{w}^T \nabla^2 \ell_i(\mathbf{w}^*) \mathbf{w}) = \nabla^2 \ell_i(\mathbf{w}^*) \mathbf{w}$. Define $\mathbf{H}_i = \nabla^2 \ell_i(\mathbf{w}^*)$, which is the Hessian of $\ell_i(\cdot)$ at $\mathbf{w}^*$. Note that $\mathbf{H}_i \in \mathbb{R}^{d \times d}$ is a Positive-Semidefinitie matrix (PSD) since $\mathbf{w}^*$ is a local minimum of $\ell_i(\cdot)$ (we refer the reader to Appendix A.1 for notation and necessary background). The linearized dynamics of SGD in our setting of interest follows below.

**Definition 1.** *Linearized SGD Dynamics: Let $\{\mathbf{H}_i\}_{i \in [n]}$ be a set of $d \times d$ PSD matrices, and let $\mathbf{H} = \frac{1}{n} \sum_{i=1}^{n} \mathbf{H}_i$. Let $\eta > 0$ denote the learning rate and $B \in [n]$ be the batch size. The linearized SGD dynamics are defined by the recurrence relation:*

$$\mathbf{w}_{t+1} = \left( \mathbf{I} - \frac{\eta}{B} \sum_{i \in \mathcal{S}} \mathbf{H}_i \right) \mathbf{w}_t, \tag{1}$$

*where $i \in \mathcal{S}$ with probability $\frac{B}{n}$ and the event $i \in \mathcal{S}$ is independent from the event $j \in \mathcal{S}$ for all $i \neq j$. We will refer to $\mathbf{J} = \mathbf{I} - \eta \mathbf{H}$ and $\hat{\mathbf{J}} \sim \mathbf{I} - \frac{\eta}{B} \sum_{i \in \mathcal{S}} \mathbf{H}_i$ as the Jacobians of GD and SGD respectively. Note that using $B = n$ recovers the gradient descent dynamics.*

---

[1] Note that linear stability is not meaningful for pure cross-entropy loss on perfectly fit data, since $\|\mathbf{w}^*\|_2$ is not finite. However, our theory holds when using label smoothing (Szegedy et al., 2016), as commonly done in practice.

# 3 THE ROLE OF HESSIAN GEOMETRY IN SGD INSTABILITY

This section introduces and motivates the Hessian coherence measure $\sigma(\{\mathbf{H}_i\}_{i\in[n]})$. Subsequently, we utilize this measure to present our primary result, Theorem 1. This theorem furnishes a sufficient condition for $\mathbb{E}\|\mathbf{w}_k\|_2$ to diverge as $k \to \infty$. Following this, we demonstrate that our established sufficient condition in Theorem 1 is nearly optimal across a broad range of hyperparameters. We formally state this optimality result in Theorem 2.

## 3.1 HESSIAN COHERENCE MEASURE

Note that, to understand the behavior of linearized SGD dynamics around a point $\mathbf{w}^*$, it suffices to consider how they operate under various configurations of $\eta$, $B$, and $\{\mathbf{H}_i\}_{i\in[n]}$. To illustrate the effect of $\{\mathbf{H}_i\}_{i\in[n]}$, let us explore two extreme scenarios.

**First Setting:** Suppose we have $\mathbf{H}_i = \mathbf{e}_1\mathbf{e}_1^T \,\forall i$, where $\mathbf{e}_i$ represents the $i$-th canonical basis vector. In this scenario, with all $\mathbf{H}_i$ being identical, we anticipate that the stochastic dynamics will closely resemble the deterministic dynamics. This expectation arises from the fact that $\frac{1}{B}\sum_{i\in\mathcal{S}}\mathbf{H}_i$ should exhibit strong concentration around $\mathbf{H}$. Furthermore, when the elements of $\mathcal{S}$ are sampled from $[n]$ without replacement, the SGD dynamics coincide with the GD dynamics. Therefore, we expect no difference in the characterization of stability of the respective linearized dynamics.

**Second Setting:** Now, let us consider the opposite extreme, where all $\mathbf{H}_i$ matrices are orthogonal, meaning that their inner products satisfy $\mathrm{tr}[\mathbf{H}_i\mathbf{H}_j] = 0 \,\forall i \neq j$. In this scenario, we anticipate that randomness exerts a substantial influence on the steps taken by SGD. In the context of a full linear GD step, the component projected onto the subspace defined by $\mathbf{H}_i$ is $\eta\mathbf{H}_i\mathbf{w}/n$. However, in the stochastic setting, if $\mathbf{H}_i$ is not selected in the sampling process, no step is taken in this particular subspace. Conversely, if it is selected, the step taken is $\eta\mathbf{H}_i\mathbf{w}/B$, which significantly overshoots the deterministic step by a factor of $n/B$.

These extreme cases serve as illustrative examples, highlighting the importance of the relative geometric arrangement within the set $\{\mathbf{H}_i\}_{i\in[n]}$ in determining the stability of the dynamics (alongside the learning rate and batch size). While we establish both sufficient and necessary conditions to address this geometric aspect in Lemma 4.1, our aim is also to offer an intuitive characterization that captures the significance of $\{\mathbf{H}_i\}_{i\in[n]}$ without resorting to complex analytical expressions. To that end, we introduce the following measure, which succinctly captures the geometric structure within $\{\mathbf{H}_i\}_{i\in[n]}$.

**Definition 2.** *Coherence Measure: For a given set of PSD matrices $\{\mathbf{H}_i\}_{i\in[n]}$, define $\mathbf{S} \in \mathbb{R}^{n\times n}$ such that $\mathbf{S}_{ij} = \|\mathbf{H}_i^{1/2}\mathbf{H}_j^{1/2}\|_F$. Equivalently, we may define $\mathbf{S}$ as the entry-wise square root of the Gram matrix of $\{\mathbf{H}_i\}_{i\in[n]}$ under the trace inner product. The coherence measure $\sigma$ is then defined as:*

$$\sigma = \frac{\lambda_1(\mathbf{S})}{\max_{i\in[n]}\lambda_1(\mathbf{H}_i)}.$$

To provide some insight into this measure, we can consider $\{\mathbf{H}_i\}_{i\in[n]}$ as a collection of $n$ vectors in $\mathbb{R}^{d\times d}$, endowed with the trace inner-product. The Gram matrix of a set of vectors compiles the pairwise inner-products among the vectors, thus representing the relative alignments and magnitudes of these vectors within the space, and the matrix $\mathbf{S}$ is an entry-wise renormalization of the Gram matrix. In the case where $\mathrm{rank}(\mathbf{H}_i) = 1$ for all $i \in [n]$, $\lambda_1(\mathbf{H}_i)$ is the $i$-th diagonal entry of $\mathbf{S}$, and $\sigma$ measure how close $\mathbf{S}$ is to being diagonally dominant. Due to the construction of $\mathbf{S}$, $\sigma$ then measures the cross-interactions within $\{\mathbf{H}_i\}_{i\in[n]}$ relative to the magnitude of the individual Hessians in $\{\mathbf{H}_i\}_{i\in[n]}$ under the Frobenius norm. The case where all $\mathbf{H}_i$ are rank one is particularly important since it occurs when $\nabla\ell(\mathbf{w}^*) = \mathbf{0}$ under loss functions such as cross-entropy loss.

Let us examine the two extreme cases mentioned earlier. In the first case, where $\mathbf{H}_i = \mathbf{e}_1\mathbf{e}_1^T \,\forall i$, it follows that $\|\mathbf{H}_i^{1/2}\mathbf{H}_j^{1/2}\|_F = 1 \,\forall i,j \in [n]$. Consequently, $\mathbf{S}$ becomes an $n \times n$ matrix consisting of all ones, yielding $\lambda_1(\mathbf{S}) = \sqrt{n}$. Meanwhile, in the scenario, where $\mathbf{H}_i$ matrices are mutually orthogonal, $\|\mathbf{H}_i^{1/2}\mathbf{H}_j^{1/2}\|_F = 0 \,\forall i \neq j$. Therefore, $\mathbf{S}$ becomes the identity matrix $\mathbf{I}$, and $\lambda_1(\mathbf{S}) = 1$. In both cases, $\lambda_1(\mathbf{H}_i) = 1$ for all $i \in [n]$. Consequently, in the first case, $\sigma = \sqrt{n}$, and in the

second case, $\sigma = 1$. This demonstrates that our coherence measure, $\sigma$, effectively distinguishes between these two extreme scenarios and increases as the alignment among the $\mathbf{H}_i$ matrices grows stronger. Below, we show how this measure allows us to establish a natural sufficient condition for the divergence of these linear dynamics.

## 3.2 SIMPLIFIED DIVERGENCE CONDITION

We present our sufficient condition for the linear dynamics to diverge, which relies solely on the values of $\lambda_1(\mathbf{H})$, $\eta$, $B$, $n$, and our coherence measure $\sigma$. Note that this bound aligns with our intuitive expectations based on the extreme cases we examine. When all $\mathbf{H}_i = \mathbf{e}_1 \mathbf{e}_1^T$ and $\sigma = \sqrt{n}$, the second condition in the theorem cannot be met. In such cases, we must resort to the GD condition for instability, namely, $\lambda_1(\mathbf{H}) > \frac{2}{\eta}$. Conversely, when $\mathbf{H}_i = \mathbf{e}_i \mathbf{e}_i^T \; \forall i$ and $\sigma = 1$, the theorem asserts that the linear dynamics will diverge even for small values of $\lambda_1(\mathbf{H})$, especially when $B$ is small relative to $n$. This behavior aligns with our expectations based on the different scenarios we consider.

**Theorem 1.** *Let $\{\hat{\mathbf{J}}_i\}_{i \in \mathbb{N}}$ be a sequence of i.i.d. copies of $\hat{\mathbf{J}}$ defined in Definition 1. Let $\{\mathbf{H}_i\}_{i \in [n]}$ have coherence measure $\sigma$. If,*

$$\lambda_1(\mathbf{H}) > \frac{2}{\eta} \;\; or \;\; \lambda_1(\mathbf{H}) > \frac{\sigma}{\eta} \cdot \left(\frac{n}{B} - 1\right)^{-1/2}, \;\; then, \;\; \lim_{k \to \infty} \mathbb{E}\|\hat{\mathbf{J}}_k...\hat{\mathbf{J}}_2 \hat{\mathbf{J}}_1\|_2 = \infty.$$

We defer all proofs to Appendix A. Note that the quantity $\mathbb{E}\|\hat{\mathbf{J}}_k...\hat{\mathbf{J}}_1\|_2 = \mathbb{E} \max_{\mathbf{w}_0: \|\mathbf{w}_0\|_2 = 1} \|\mathbf{w}_k\|_2$, where $\mathbf{w}_k$ is the random vector determined by the SGD dynamics in Definition 1, when the dynamics start from $\mathbf{w}_0$. In other words, if $\mathbb{E}\|\hat{\mathbf{J}}_k...\hat{\mathbf{J}}_1\|_2$, diverges as $k \to \infty$, the linearized SGD dynamics diverge from almost every starting point $\mathbf{w}_0 \in \mathbb{R}^d$. We highlight two observations of the condition in Theorem 1. First, this analysis supports "squared-scaling" between batch size and learning rate, that is, if $B$ is increased proportional to $\eta^2$, the stability will not increase. Second, the Hessian alignment (captured by $\sigma$) can cause instability even when $\eta$ is small and $B$ is on the order of $n$.

### 3.2.1 COMPARISON TO PRIOR WORK

Based on the formal introduction of our result, we now provide a more detailed comparison to the result of Wu et al. (2022). The condition outlined in Theorem 3.3 of Wu et al. (2022) represents one of the most recent findings in this research field, which we restate in Appendix B. The contrapositive of this theorem provides a sufficient condition, namely $\|\mathbf{H}\|_F > \frac{1}{\eta}\sqrt{\frac{B}{\mu_0}}$, to guarantee instability. Here, $\mu_0$ serves as an alignment metric, empirically argued to have a practical lower bound.

Theorem 1 has multiple advantages over these prior results. First, the analysis in Wu et al. (2022) is confined solely to the MSE loss function. Second, their definition of stability entails that $\mathbb{E}[L(\mathbf{w}_k)] \leq C \cdot \mathbb{E}[L(\mathbf{w}_1)] \; \forall k \in \mathbb{N}$, where $C > 1$ is a constant. This definition is notably weaker than our notion of stability. Finally, despite Theorem 1 holding for more general loss functions and guaranteeing a stronger notion of stability/instability, it is also easier to satisfy under the typical setting where $\mathbf{H}$ has low stable rank, i.e., when $\|\mathbf{H}\|_F^2/\|\mathbf{H}\|_2^2$ is small. Let us ignore the effects of the measures $\mu_0$ and $\sigma$ by considering both equal to one, then Theorem 3.3 (Wu et al., 2022) guarantees instability if $\|\mathbf{H}\|_F > \frac{1}{\eta}\sqrt{B}$ while Theorem 1 guarantees instability if $\lambda_1(\mathbf{H}) > \frac{1}{\eta}\sqrt{\frac{B}{n}}$, which is a more general condition when the stable rank of $\mathbf{H}$ is less than $n$, as is typical in practical settings (Xie et al., 2022).

## 3.3 OPTIMALITY OF THEOREM 1

Theorem 1 provides a sufficient condition for the linear dynamics to diverge, where the condition is of the form $\lambda_1(\mathbf{H}) > f(\eta, \sigma, n, B)$, where $f(\eta, \sigma, n, B) = \frac{\sigma}{\eta}\left(\frac{n}{B} - 1\right)^{-1/2}$. The next theorem shows that our condition is optimal in the sense that, for a natural range of parameters (when $\sigma, \frac{n}{B} = \mathcal{O}(1)$), the function $f(\eta, \sigma, n, B)$ is within a constant factor of its lowest possible value. This shows that our sufficient condition cannot be significantly relaxed without relying on other information about the set $\{\mathbf{H}_i\}$ as we do in Lemma 4.1.

Overall, the following theorem demonstrates that our sufficient condition for divergent dynamics approaches optimality among all sufficient conditions that rely solely on $\eta$, $\sigma$, $B$, $\lambda_1(\mathbf{H})$, and $n$. However, it does not rule out further improvement in the important regime where $B \ll n$.

**Theorem 2.** *For every choice of $\lambda_1 > 0$, $n \in \mathbb{N}$, $B \in [n]$, $\eta > 0$, and $\sigma \in [n]$, that satisfies:*

$$\lambda_1 < \frac{2\sigma}{\eta} \cdot \left(\sigma + \frac{n}{B} - 1\right)^{-1},$$

*There exists a set of PSD matrices $\{\mathbf{H}_i\}_{i \in [n]}$ such that $\lambda_1(\mathbf{H}) = \lambda_1$ and $\lim_{k \to \infty} \mathbb{E}\|\hat{\mathbf{J}}_k...\hat{\mathbf{J}}_1\|_F^2 < n$.*

## 4  SHARP STABILITY CONDITIONS OF LINEARIZED SGD

In the previous section, we provide a measure of the geometric coherence in $\{\mathbf{H}_i\}_{i \in [n]}$ along with a theorem that provides sufficient conditions for the dynamics of SGD (Definition 1) to diverge. The proof of this theorem relies on part (i) of the following technical lemma. Aside from its utility in establishing the aforementioned theorem, the statement of the following lemma also imparts valuable insights into the behavior of linearized SGD dynamics.

The proof of the following lemma relies on the observation that $\|\mathbf{M}\|_2^2 \leq \|\mathbf{M}\|_F^2 \leq d\|\mathbf{M}\|_2^2$ for all $\mathbf{M} \in \mathbb{R}^{d \times d}$, where $\|\cdot\|_F$ denotes the Frobenius norm. Hence, we may focus on divergence in the Frobenius norm of the $k$-step linearized dynamics. Now, $\mathbb{E}\|\hat{\mathbf{J}}_k...\hat{\mathbf{J}}_1\|_F^2 = \mathbb{E}[\text{tr}[\hat{\mathbf{J}}_k...\hat{\mathbf{J}}_1^2...\hat{\mathbf{J}}_k]] = \text{tr}[\mathbb{E}[\hat{\mathbf{J}}_k...\hat{\mathbf{J}}_1^2...\hat{\mathbf{J}}_k]]$ by linearity. By operator monotonicity of the trace, we further have $\text{tr}[\mathbf{N}_k] \geq \text{tr}[\mathbb{E}[\hat{\mathbf{J}}_k...\hat{\mathbf{J}}_1^2...\hat{\mathbf{J}}_k]] \geq \text{tr}[\mathbf{M}_k]$, where $\mathbf{N}_k \succeq \mathbb{E}[\hat{\mathbf{J}}_k...\hat{\mathbf{J}}_1^2...\hat{\mathbf{J}}_k] \succeq \mathbf{M}_k$ under the Loewner ordering (see Appendix A.1). The technical challenge in our proof lies in composing an inductive argument to define matrices $\mathbf{N}_k$ and $\mathbf{M}_k$. We opt for an approach that directly bounds the matrix of the $k$-step linear dynamics under the Loewner ordering, which does introduce greater technical complexity compared to using norm inequalities, as seen in previous work. However, this added complexity is essential to accurately account for the alignment in the unstable eigenvectors of each $\hat{\mathbf{J}}_i$, allowing us to provide a thorough characterization of the instability of SGD dynamics.

**Lemma 4.1.** *Let $\hat{\mathbf{J}}_i$ be independent Jacobians of SGD dynamics described in Definition 1.*

*(i) If*

$$\lambda_1(\mathbf{H}) > \frac{2}{\eta} \quad or \quad \lim_{k \to \infty} \left(\frac{\eta^2}{nB} - \frac{\eta^2}{n^2}\right)^k \sum_{y_1...,y_k=1}^n \|\mathbf{H}_{y_k}...\mathbf{H}_{y_1}\|_F^2 = \infty,$$

*then $\lim_{k \to \infty} \mathbb{E}\|\hat{\mathbf{J}}_k...\hat{\mathbf{J}}_1\|_F^2 = \infty$.*

*(ii) If, for some $\epsilon \in (0, 1)$,*

$$\frac{\epsilon}{\eta} < \lambda_i(\mathbf{H}) < \frac{2 - \epsilon}{\eta} \text{ for all } i \in [d] \quad and \quad \lim_{k \to \infty} \frac{1}{\epsilon^k} \left(\frac{\eta^2}{nB} - \frac{\eta^2}{n^2}\right)^k \sum_{y_1...,y_k=1}^n \|\mathbf{H}_{y_k}...\mathbf{H}_{y_1}\|_F^2 = 0,$$

*then $\lim_{k \to \infty} \mathbb{E}\|\hat{\mathbf{J}}_k...\hat{\mathbf{J}}_1\|_F^2 = 0$.*

Notice that part (i) and part (ii) of this theorem are complementary in the sense that the condition of part (ii) is nearly the negation of the condition in part (i) except for the additional $\epsilon$ factor. In a sense, the parameter $\epsilon$ captures the balance of how close we are to instability in GD dynamics, i.e. $\lambda_1(\mathbf{H}) > \frac{2}{\eta}$, and how much additional instability is added by the stochasticity in the dynamics of SGD. [2]

To provide a more detailed explanation of this intuition, let us consider the setting where $B \ll n$. The second term from part (ii) of the above Lemma can be approximated as:

$$\left(\frac{\eta^2}{nB} - \frac{\eta^2}{n^2}\right)^k \sum_{y_1...,y_k=1}^n \|\mathbf{H}_{y_k}...\mathbf{H}_{y_1}\|_F^2 \approx \frac{\eta^{2k}}{n^k B^k} \sum_{y_1...,y_k=1}^n \|\mathbf{H}_{y_k}...\mathbf{H}_{y_1}\|_F^2 = \frac{\eta^{2k}}{B^k} \cdot \mathbb{E}\|\mathbf{A}_k...\mathbf{A}_1\|_F^2,$$

$$(2)$$

---

[2]We believe that the requirement in part (ii) that $\lambda_i(\mathbf{H}) > \frac{\epsilon}{\eta}$ could likely be removed by more carefully accounting for alignment of the negligible eigenvectors of $\hat{\mathbf{J}}$ and $\mathbf{J}$ and relaxing the theorem to imply boundedness of the limit. However, given that the role of part (ii) in the theorem is only to contrast with part (i), we do not think this is high priority for the purpose of this paper.

where $\mathbf{A}_i$ is independently sampled uniformly from the set $\{\mathbf{H}_i\}_{i\in[n]}$. Interestingly, this implies that the effect of the parameters $B$ and $\eta$ can be decoupled from the structure within $\{\mathbf{H}_i\}_{i\in[n]}$. In other words, if we knew the minimal learning rate $\eta$ at which linearized SGD with a fixed batch size diverges, then we would immediately be able to determine which parameter pairs $(\eta, B)$ are divergent at the given point $\mathbf{w}^*$, since the term $\mathbb{E}\|\mathbf{A}_k...\mathbf{A}_1\|_F^2$ does not change with these hyperparameters.

Note that determining whether the quantity $\frac{\eta^{2k}}{B^k}\mathbb{E}\|\mathbf{A}_k...\mathbf{A}_1\|_F^2$ diverges for arbitrary inputs of $\eta$, $B$, and set of arbitrary symmetric matrices $\{\mathbf{H}_i\}_{i\in[n]}$ would be an NP-Hard problem (see Section 3.3 of Huang et al. (2022)). Even in our case, where we have the additional constraint that each $\mathbf{H}_i$ is PSD, we are unaware of an efficient method to determine whether $\frac{\eta^{2k}}{B^k}\mathbb{E}\|\mathbf{A}_k...\mathbf{A}_1\|_F^2$ diverges, motivating our simplified sufficient condition in Theorem 1.

## 5 EXPERIMENTS

In this section, we support our prior theorems by empirically evaluating the behavior of SGD on synthetic optimization problems with additively decomposable loss functions. The high-level points our experiments support are:

- Parameter tuples that Theorem 1 guarantees divergence do indeed diverge.
- Parameter tuples that Theorem 2 guarantees no divergence indeed do not diverge.
- The coherence measure $\sigma$ has an important effect on the instability of SGD.
- Our theoretical results hold when using SGD that samples without replacement.

The first two points outlined above serve as validation for the accuracy and soundness of our theoretical results and proofs. The third point enhances the rationale for adopting the Hessian coherence measure we introduce. Lastly, the fourth point offers justification for employing SGD with Bernoulli sampling in our theoretical analysis, as its behavior mirrors that of the more prevalent SGD approach that samples without replacement. To ensure reproducibility, we include all our implementations in the supplementary material.

### 5.1 EXPERIMENT SETUP

We leverage the construction used in the proof of Theorem 2 to verify our predictions empirically, which offers two advantages: 1) we may apply the analysis of Theorem 2 for a condition that guarantees no divergence, and 2) the construction is parameterized by $\sigma$, and so we may easily test the effect of varying $\sigma$. In this construction, we set $\mathbf{H}_i = m \cdot \mathbf{e}_1\mathbf{e}_1^T$ for all $i \in [\sigma]$ and $\mathbf{H}_i = m \cdot \mathbf{e}_{i-\sigma+1}\mathbf{e}_{i-\sigma+1}^T$ otherwise, with $m = \frac{2n}{\sigma}$. We set the dimension of the space to $n-\sigma+1$, so there is a unique minimizer of the loss, as this does not affect divergence. Notice that this construction essentially interpolates the two extreme settings in Section 3.1 as $\sigma$ varies from $\sigma = 1$ to $\sigma = n$. Additionally, note that $\lambda_1(\mathbf{H}) = 2$ by construction. In our experiments, $\eta \leq 1$, hence the first condition of Theorem 1, i.e., $\lambda_1(\mathbf{H}) > 2/\eta$ will not hold or be relevant to characterizing stability.

The loss function that corresponds to the set of Hessians $\{\mathbf{H}_i\}_{i\in[n]}$ is given by the additively decomposable quadratic function $L(\mathbf{w}) = \frac{1}{n}\sum_{i=1}^n \ell_i(\mathbf{w})$, where $\ell_i(\mathbf{w}) = \mathbf{w}^T\mathbf{H}_i\mathbf{w}$. Note that, for this construction of $\{\mathbf{H}_i\}_{i\in[n]}$, $\ell_i(\mathbf{w})$ is particularly easy to compute, as it is equivalent to squaring and rescaling a single entry of $\mathbf{w}$.

Across all experiments, we set $n = 100$. For each set of parameters $(B, \eta, \sigma)$, we determine whether the combination leads to divergence or not by executing SGD for a maximum of 1000 steps. Specifically, we classify a tuple as divergent if, in the majority of the five repetitions, the norm of the parameter vector $\mathbf{w}$ increases by a factor of 1000. Conversely, we terminate SGD prematurely and classify the point as not divergent if the norm of $\mathbf{w}$ decreases by a factor of 1000 during the course of the SGD trajectory.

It is possible to construct a regression problem with mean-squared error that results in exactly the same optimization we describe. However, we note that Theorem 3.3 in Wu et al. (2022) does not directly apply to this problem, since the corresponding loss is zero at the optimum, and hence the loss-scaled alignment factor (see Equation 4 below) of Wu et al. (2022) is undefined at the optimum. Therefore, the condition of Theorem 3.3 in Wu et al. (2022) (see Theorem 3) cannot be satisfied.

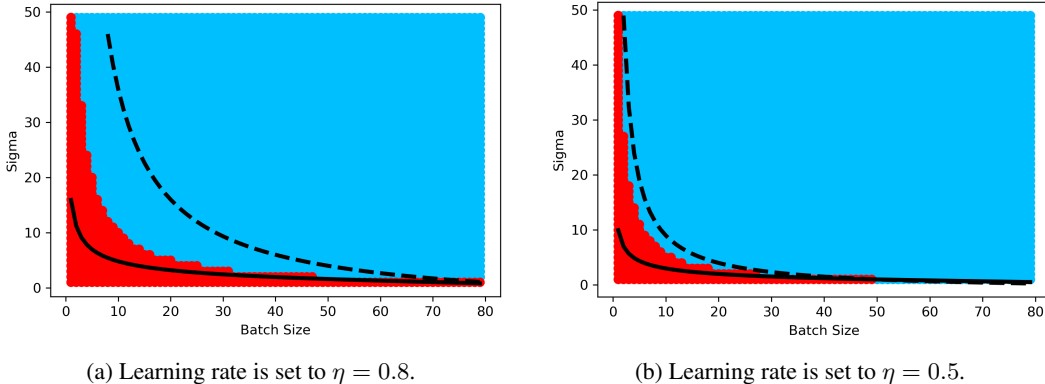

(a) Learning rate is set to $\eta = 0.8$.                    (b) Learning rate is set to $\eta = 0.5$.

Figure 1: The red area indicates where SGD diverges and blue where it does not diverge among parameter pairs $(\sigma, B)$. The solid black line is where the condition of Theorem 1 attains equality and the dashed line is where the condition of Theorem 2 attains equality.

## 5.2 EXPERIMENTAL RESULTS

### 5.2.1 EFFECT OF COHERENCE MEASURE AND BATCH SIZE

First, we look at how the stability of SGD changes as we vary coherence measure $\sigma$ and batch size $B$. We show results for two fixed values of the learning rate, $\eta = 0.8$ and $\eta = 0.5$.

Two key observations in Figure 1 are that all tuples $(\sigma, B)$ that are below the boundary given by Theorem 1 indeed diverge. This is visually shown as all points below the solid black line are red (besides some aberration due to visual smoothing). Additionally, the fact that all points above the dashed line are blue indicate that tuples $(\sigma, B)$ which the proof of Theorem 2 guarantees converge indeed converge.

An intriguing observation is the pattern where the gap between the upper and lower bounds diminishes as the batch size increases. Specifically, we notice that the lower bound more closely aligns with the actual boundary between divergence and convergence across all batch sizes when $\eta = 0.5$. However, the upper bound is closer to the true boundary when $\eta = 0.8$.

Finally, we observe that the coherence measure $\sigma$ exerts a substantial influence on the stability of SGD. For small values of $\sigma$, SGD demonstrates instability even at high values of $B$. This observation underscores the importance of considering the geometry of the loss surface in understanding the behavior of SGD. Furthermore, it highlights that the coherence measure is an effective tool for capturing and accounting for the contribution of loss surface geometry to the stability of SGD.

### 5.2.2 EFFECT OF BATCH SIZE AND LEARNING RATE

Next, we examine how the stability of SGD evolves when we manipulate the batch size $B$ and the learning rate $\eta$. For this analysis, we maintain a fixed value of $\sigma = 5$, which provides a clear boundary given the granularity of the point grid. We show both the log-scale plot, since learning rate generally varies on a log-scale, and the linear-scale plot since we expect the relationship between $B$ and $\eta^2$ to be roughly linear in the boundary.

The plots displayed in Figure 2 further corroborate the validity of Theorem 1 and Theorem 2. Additionally, the pattern continues to support the facts that the lower bound condition more closely approximates the true boundary as learning rate decreases, and the upper bound provides a tighter approximation to the true boundary as learning rate increases.

## 6 CONCLUSION

We present precise yet interpretable, necessary and sufficient conditions to determine when a point $\mathbf{w}^*$ is stable under linearized SGD dynamics. The sufficient condition in Theorem 1 relies on a novel

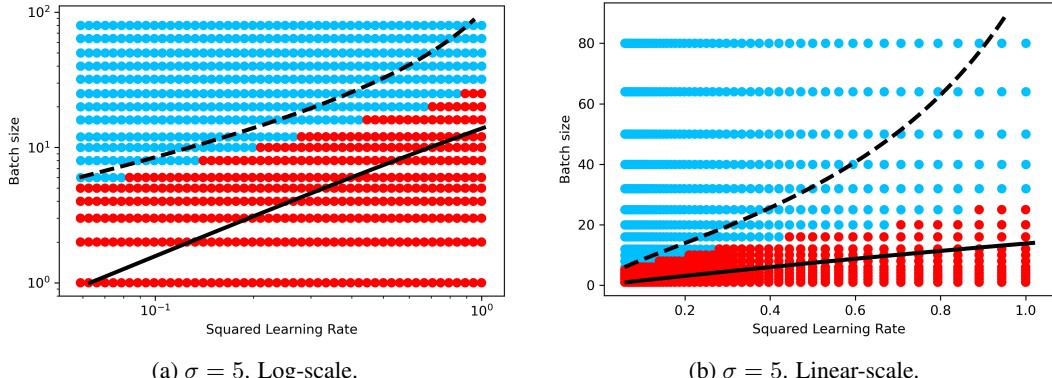

Figure 2: The red area indicates where SGD diverges and grey where it does not diverge among parameter pairs $(\eta, B)$. We plot the squared value of $\eta^2$ to make the linear relation between $B$ and $\eta^2$ clearer. The solid black line is where the condition of Theorem 1 attains equality and the dashed line is where the condition of Theorem 2 attains equality.

coherence measure $\sigma$ that summarizes relevant information in the loss surface geometry. We next list some open questions our work raises:

- In future research endeavors, it would be intriguing to close the gap between Theorem 1 and Theorem 2 to establish the actual dependency of SGD stability on $\sigma$ and the hyperparameters of SGD.

- Additionally, it would be interesting to empirically measure the value of the coherence measure $\sigma$ in realistic neural networks. Acquiring knowledge about the practical range of values that $\sigma$ can assume would enhance the utility of the theoretical contributions provided here for predicting the behavior of SGD in real-world scenarios. Developing efficient approaches to approximate $\sigma$ in large neural networks would represent a valuable step toward achieving this objective.

- We may also consider extending the same proof techniques to characterize the stability of sharpness-aware methods (Behdin et al., 2023; Foret et al., 2020; Zhuang et al., 2022; Liu et al., 2022; Kwon et al., 2021; Kim et al., 2022), which are commonly employed for training many overparameterized models.

- Along these lines, it would also be useful to consider whether the stability of SGD with momentum or other adaptive gradient methods could be analyzed with this approach.

- The convergence analysis in Lemma 4.1 could possibly used to derive fine-grained local convergence rates of SGD depending on Hessian alignment.

## ACKNOWLEDGMENTS

GD was partially supported by NSF AF 1814041, NSF FRG 1760353, and DOE-SC0022085. RK would like to acknowledge support from AnalytiXIN Indiana.

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

# A PROOFS

## A.1 PROOF PRELIMINARIES

In this section, we provide notation and necessary background for the following proofs.

**Notation:** Let $[n]$ denote $1, 2, ..., n$. Let $\mathbf{e}_i$ denote $i$-th canonical basis vector. Let $\text{Bern}(p)$ be Bernoulli distribution with parameter $p$. Let $\mathbf{0}$ denote the all-zero matrix or vector, where the dimension will be clear from context. Let $\mathbf{1}$ be defined similarly as the all-ones matrix or vector. Let $\|\mathbf{A}\|_2$ denote the spectral norm of matrix $\mathbf{A}$. Let $\lambda_i(\mathbf{A})$ denote the $i$-th largest eigenvalue of the matrix $\mathbf{A}$. Let $\|\mathbf{A}\|_F$ denote the Frobenius norm of matrix $\mathbf{A}$. Let $\|\mathbf{A}\|_{\mathcal{S}_p}$ denote $p$-Schatten norm of matrix $\mathbf{A}$, that is, the $p$-norm of the vector of singular values of $\mathbf{A}$.

**Fact 1.** $\ell_1$-$\ell_2$ *Norm Inequality: For any vector* $\mathbf{x} \in \mathbb{R}^d$, $\|\mathbf{x}\|_2 \le \|\mathbf{x}\|_1 \le \sqrt{d}\|\mathbf{x}\|_2$.

**Fact 2.** *Recursive Formula for Binomial Coefficients: For all* $n, k \in \mathbb{N}$ *such that* $k \le n$, *the binomial coefficients satisfy the following recursive formula:*

$$\binom{n}{k} = \binom{n-1}{k-1} + \binom{n-1}{k}.$$

The notion of PSD matrices can be used to define a partial order on the set of symmetric matrices as follows.

**Definition 3.** *Loewner Order: The Loewner order is a partial order on the set of positive semidefinite symmetric matrices. For two positive semidefinite matrices* $\mathbf{A}$ *and* $\mathbf{B}$, *we write* $\mathbf{A} \preceq \mathbf{B}$ *to denote that* $\mathbf{B} - \mathbf{A}$ *is positive semidefinite and* $\mathbf{A} \prec \mathbf{B}$ *to denote that* $\mathbf{B} - \mathbf{A}$ *is positive definite.*

Note that if $\mathbf{A}$ is a PSD matrix, then for any symmetric $\mathbf{B}$, $\mathbf{B}\mathbf{A}\mathbf{B}$ must also be PSD.

We frequently use the following properties of the trace in our derivations.

**Properties of the Trace:**

- $\text{tr}[\mathbf{A}] = \sum_{i=1}^n \lambda_i(\mathbf{A})$.

- Invariance to cyclic permutation: $\text{tr}[\mathbf{A}\mathbf{B}\mathbf{C}] = \text{tr}[\mathbf{C}\mathbf{A}\mathbf{B}]$.

- Linearity of the trace: $\text{tr}[c\mathbf{A} + \mathbf{B}] = c\,\text{tr}[\mathbf{A}] + \text{tr}[\mathbf{B}]$, where $c \in \mathbb{R}$.

**Lemma A.1.** *For any matrix* $\mathbf{M} \in \mathbb{R}^{n \times n}$, $\|\mathbf{M}\|_F \le \|\mathbf{M}\|_{\mathcal{S}_1} \le \sqrt{n}\|\mathbf{M}\|_F$.

*Proof.* This follows from the fact that $\|\mathbf{M}\|_F = \|\mathbf{M}\|_{\mathcal{S}_2}$ and $\|\mathbf{M}\|_{\mathcal{S}_p}$ is the $p$-norm of the spectrum of $\mathbf{M}$ along with applying the $\ell_1$-$\ell_2$-norm inequality (Fact 1). $\square$

**Lemma A.2.** *For any length* $k$ *sequence of square matrices* $\mathbf{A}_1...\mathbf{A}_k \in \mathbb{R}^{d \times d}$, $\text{tr}[\mathbf{A}_1\mathbf{A}_2...\mathbf{A}_k] \le \sqrt{d}\|\mathbf{A}_1\|_F\|\mathbf{A}_2\|_F...\|\mathbf{A}_k\|_F$.

*Proof.* First, it follows from Weyl's Majorant Theorem that $\text{tr}[\mathbf{A}_1\mathbf{A}_2...\mathbf{A}_k] \le \|\mathbf{A}_1\mathbf{A}_2...\mathbf{A}_k\|_{\mathcal{S}_1}$ (see Section III.5 (Bhatia, 2013)). Then, applying Lemma A.1 implies,

$$\text{tr}[\mathbf{A}_1\mathbf{A}_2...\mathbf{A}_k] \le \|\mathbf{A}_1\mathbf{A}_2...\mathbf{A}_k\|_{\mathcal{S}_1} \le \sqrt{d}\|\mathbf{A}_1\mathbf{A}_2...\mathbf{A}_k\|_F.$$

Finally, we conclude the lemma statment by submultiplicativity of the Frobenius norm. $\square$

## A.2 PROOFS

**Proof of Theorem 1**

*Proof.* Note that $\|\mathbf{M}\|_2^2 \le \|\mathbf{M}\|_F^2 \le d\|\mathbf{M}\|_2^2$ for all $\mathbf{M} \in \mathbb{R}^{d \times d}$. Therefore, by Lemma 4.1, it suffices to show that the condition of the theorem implies that the following quantity diverges towards infinity as $k \to \infty$:

$$\left(\frac{\eta^2}{nB} - \frac{\eta^2}{n^2}\right)^k \sum_{y_1...,y_k=1}^n \|\mathbf{H}_{y_k}...\mathbf{H}_{y_1}\|_F^2$$

From here, we lower bound it by the following. Note that $\|\mathbf{M}\|_F^2 \geq \frac{1}{d}\|\mathbf{M}\|_{\mathcal{S}_1}^2$, where $\|\mathbf{M}\|_{\mathcal{S}_1}$ is the 1-Schatten norm (see Lemma A.1).

$$
\begin{aligned}
\left(\frac{\eta^2}{nB} - \frac{\eta^2}{n^2}\right)^k \sum_{y_1\ldots,y_k=1}^n \|\mathbf{H}_{y_k}...\mathbf{H}_{y_1}\|_F^2 &\geq \left(\frac{\eta^2}{nB} - \frac{\eta^2}{n^2}\right)^k \sum_{y_1\ldots,y_k=1}^n \frac{1}{d}\|\mathbf{H}_{y_k}...\mathbf{H}_{y_1}\|_{\mathcal{S}_1}^2 \\
&\geq \left(\frac{\eta^2}{nB} - \frac{\eta^2}{n^2}\right)^k \sum_{y_1\ldots,y_k=1}^n \frac{1}{d}\operatorname{tr}[\mathbf{H}_{y_k}...\mathbf{H}_{y_1}]^2 \\
&\geq \left(\frac{\eta^2}{nB} - \frac{\eta^2}{n^2}\right)^k \frac{1}{d}\sum_{y=1}^n \operatorname{tr}[\mathbf{H}_y^k]^2 \\
&\geq \left(\frac{\eta^2}{nB} - \frac{\eta^2}{n^2}\right)^k \frac{1}{nd}\left(\sum_{y=1}^n \operatorname{tr}[\mathbf{H}_y^k]\right)^2
\end{aligned}
$$

The last line follows by again applying the L1-L2 norm inequality. Now we show that the complexity measure $\sigma$ satisfies:

$$
\frac{n^k}{d^2 \cdot \sigma^k} \cdot \operatorname{tr}[\mathbf{H}^k] \leq \sum_{y=1}^n \operatorname{tr}[\mathbf{H}_y^k].
$$

First, we bound the left-hand term as follows. By Lemma A.2 we know that $\operatorname{tr}[\mathbf{A}_1\mathbf{A}_2...\mathbf{A}_k] \leq d\|\mathbf{A}_1\|_F\|\mathbf{A}_2\|_F...\|\mathbf{A}_{k-1}\|_F\|\mathbf{A}_k\|_F$. Therefore,

$$
\begin{aligned}
\operatorname{tr}[\mathbf{H}^k] &= \frac{1}{n^k}\cdot\sum_{y_1\ldots,y_k=1}^n \operatorname{tr}[\mathbf{H}_{y_k}...\mathbf{H}_{y_1}] \\
&= \frac{1}{n^k}\cdot\sum_{y_1\ldots,y_k=1}^n \operatorname{tr}[(\mathbf{H}_{y_1}^{1/2}\mathbf{H}_{y_k}^{1/2})(\mathbf{H}_{y_k}^{1/2}\mathbf{H}_{y_{k-1}}^{1/2})...(\mathbf{H}_{y_2}^{1/2}\mathbf{H}_{y_1}^{1/2})] \\
&\leq \frac{d}{n^k}\cdot\sum_{y_1\ldots,y_k=1}^n \|\mathbf{H}_{y_1}^{1/2}\mathbf{H}_{y_k}^{1/2}\|_F \cdot \|\mathbf{H}_{y_k}^{1/2}\mathbf{H}_{y_{k-1}}^{1/2}\|_F \cdot ... \cdot \|\mathbf{H}_{y_2}^{1/2}\mathbf{H}_{y_1}^{1/2}\|_F \\
&= \frac{d}{n^k}\cdot\sum_{y_1\ldots,y_k=1}^n \mathbf{S}_{y_1,y_k}\mathbf{S}_{y_k,y_{k-1}}...\mathbf{S}_{y_2,y_1} \\
&= \frac{d}{n^k}\cdot\operatorname{tr}[\mathbf{S}^k] \leq \frac{d^2}{n^k}\cdot\lambda_1(\mathbf{S})^k.
\end{aligned}
$$

Therefore,

$$
\frac{n^k}{d^2\cdot\sigma^k}\cdot\operatorname{tr}[\mathbf{H}^k] = \frac{n^k\cdot\max_{i\in[n]}\lambda_1(\mathbf{H}_i)^k}{d^2\cdot\lambda_1(\mathbf{S})^k}\cdot\operatorname{tr}[\mathbf{H}^k] \leq \max_{i\in[n]}\lambda_1(\mathbf{H}_i)^k \leq \sum_{y=1}^n\operatorname{tr}[\mathbf{H}_y^k].
$$

Starting from where we left off before,

$$
\begin{aligned}
\mathbb{E}\operatorname{tr}[\hat{\mathbf{J}}_k...\hat{\mathbf{J}}_1^2...\hat{\mathbf{J}}_k] &\geq \left(\frac{\eta^2}{nB} - \frac{\eta^2}{n^2}\right)^k \frac{1}{nd}\left(\sum_{y=1}^n \operatorname{tr}[\mathbf{H}_y^k]\right)^2 \\
&\geq \left(\frac{\eta^2}{nB} - \frac{\eta^2}{n^2}\right)^k \frac{1}{nd}\left(\frac{n^k}{d^2\cdot\sigma^k}\cdot\operatorname{tr}[\mathbf{H}^k]\right)^2 \\
&\geq \left(\frac{\eta^2}{nB} - \frac{\eta^2}{n^2}\right)^k \frac{1}{nd^5}\cdot\frac{n^{2k}}{\sigma^{2k}}\cdot\lambda_1(\mathbf{H})^{2k} \\
&= \eta^{2k}\left(\frac{n}{\sigma^2 B} - \frac{1}{\sigma^2}\right)^k \frac{1}{nd^5}\cdot\lambda_1(\mathbf{H})^{2k}
\end{aligned}
$$

We see that by the condition of the theorem that $\lambda_1(\mathbf{H}) > \frac{1}{\eta}\cdot\left(\frac{n}{\sigma^2 B} - \frac{1}{\sigma^2}\right)^{-1/2}$ that this above equation diverges towards infinity as $k \to \infty$. Hence, we conclude the theorem statement. $\square$

**Proof of Theorem 2**

*Proof.* Construct $\{\mathbf{H}_i\}_{i \in [n]}$ so that $\mathbf{H}_i = m \cdot \mathbf{e}_1 \mathbf{e}_1^T$ if $i \in [\sigma]$ and $\mathbf{H}_i = \mathbf{0}$ otherwise, where $m = \frac{\lambda_1 \cdot n}{\sigma}$. Note that $\lambda_1(\mathbf{H}) = \frac{\sigma}{n} \cdot m = \lambda_1$. Furthermore, note that $\|\mathbf{H}_i^{1/2} \mathbf{H}_j^{1/2}\|_F = m$ if $i, j \in [\sigma]$ and zero otherwise. Therefore, $\mathbf{S}$, the entry-wise square root of the Gram matrix (i.e, $\mathbf{S}_{ij} = \|\mathbf{H}_i^{1/2}\mathbf{H}_j^{1/2}\|_F$), has all of the first $\sigma \times \sigma$ entries equal to $m$ and the rest equal to zero. Therefore, $\lambda_1(\mathbf{S}) = m \cdot \sigma$. Meanwhile, $\max_{i \in [n]} \lambda_1(\mathbf{H}_i) = m$. Hence, we see the Hessian coherence measure for this constructed problem is indeed equal to the chosen value of $\sigma$.

We now show that $\lim_{k \to \infty} \mathbb{E}\|\hat{\mathbf{J}}_k....\hat{\mathbf{J}}_1\|_F^2 < n$. Note that $\hat{\mathbf{J}} = \mathbf{I} - \eta \sum_{i=1}^n x_i \mathbf{H}_i$, where the $x_i$'s are i.i.d. random variables sampled from a Bernoulli distribution with parameter $p = B/n$. A key observation here is that all $\hat{\mathbf{J}}$ are diagonal (since $\mathbf{I}$ and all $\mathbf{H}_i$ are diagonal) and hence $\hat{\mathbf{J}}_1...\hat{\mathbf{J}}_k$ commute. Therefore,

$$\mathbb{E}\|\hat{\mathbf{J}}_k....\hat{\mathbf{J}}_1\|_F^2 = \mathbb{E}\operatorname{tr}[\hat{\mathbf{J}}_k...\hat{\mathbf{J}}_1^2...\hat{\mathbf{J}}_k] = \mathbb{E}\operatorname{tr}[\hat{\mathbf{J}}_k^2...\hat{\mathbf{J}}_1^2] = \operatorname{tr}[\mathbb{E}[\hat{\mathbf{J}}_1^2]^k],$$

where in the last step we use the fact that all of the $\hat{\mathbf{J}}$ are independent and identically distributed. Recall that the trace is the sum of the diagonal entries in a matrix and hence $\operatorname{tr}[\mathbb{E}[\hat{\mathbf{J}}_1^2]^k] = \sum_{i=1}^n \mathbf{e}_i^T \mathbb{E}[\hat{\mathbf{J}}_1^2]^k \mathbf{e}_i$. Note that $\mathbf{e}_i^T \mathbb{E}[\hat{\mathbf{J}}_1^2]^k \mathbf{e}_i = 1$ when $i \neq 1$, so $\operatorname{tr}[\mathbb{E}[\hat{\mathbf{J}}_1^2]^k] = (n-1) + \mathbf{e}_1^T \operatorname{tr}[\mathbb{E}[\hat{\mathbf{J}}_1^2]^k]\mathbf{e}_1$. To show that this quantity is bounded by $n$, we must show that $\mathbf{e}_1^T \mathbb{E}[\hat{\mathbf{J}}_1^2]^k \mathbf{e}_1 = (\mathbf{e}_1^T \mathbb{E}[\hat{\mathbf{J}}_1^2]\mathbf{e}_1)^k$ is bounded. Following the base case of part (i) in Lemma 4.1, we have that:

$$\mathbb{E}[\hat{\mathbf{J}}_1^2] = \mathbf{I} - 2\eta\mathbf{H} + \eta^2\mathbf{H}^2 + \left(\frac{\eta^2}{nB} - \frac{\eta^2}{n^2}\right)\sum_{i=1}^n \mathbf{H}_i^2$$

We can then write $\mathbf{e}_1^T \mathbb{E}[\hat{\mathbf{J}}_1^2]\mathbf{e}_1$ as:

$$\mathbf{e}_1^T \mathbb{E}[\hat{\mathbf{J}}_1^2]\mathbf{e}_1 = 1 - 2\eta\frac{\sigma}{n} \cdot m + \eta^2\frac{\sigma^2}{n^2}m^2 + \left(\frac{\eta^2}{nB} - \frac{\eta^2}{n^2}\right) \cdot \sigma \cdot m^2,$$

where we used the fact that $\mathbf{e}_1^T\mathbf{H}\mathbf{e}_1 = \lambda_1(\mathbf{H}) = \frac{\sigma}{n} \cdot m$ in our construction and $\mathbf{e}_1^T\mathbf{H}_i^2\mathbf{e}_1 = m^2$ if $i \in [\sigma]$ and zero otherwise. To show convergence, we must show that the previous equation is less than 1. This is equivalent to the following condition:

$$\eta^2\frac{\sigma^2}{n^2}m^2 + \left(\frac{\eta^2}{nB} - \frac{\eta^2}{n^2}\right) \cdot \sigma \cdot m^2 < 2\eta \cdot \frac{\sigma}{n} \cdot m$$

$$\iff \eta^2\frac{\sigma^2}{n^2}m^2 + \eta^2\frac{\left(\frac{n}{B} - 1\right)}{\sigma} \cdot \frac{\sigma^2}{n^2} \cdot m^2 < 2\eta \cdot \frac{\sigma}{n} \cdot m$$

$$\iff \eta^2\lambda_1(\mathbf{H})^2 + \eta^2\frac{\left(\frac{n}{B} - 1\right)}{\sigma} \cdot \lambda_1(\mathbf{H})^2 < 2\eta \cdot \lambda_1(\mathbf{H})$$

$$\iff \lambda_1(\mathbf{H}) < \frac{2}{\eta} \cdot \left(1 + \frac{n/B - 1}{\sigma}\right)^{-1}$$

$$\iff \lambda_1 < \frac{2\sigma}{\eta} \cdot \left(\sigma + \frac{n}{B} - 1\right)^{-1}.$$

Therefore, we conclude the theorem statement

$\square$

**Proof of Lemma 4.1**

*Proof.* First, note that $\|\mathbf{M}\|_F^2 = \operatorname{tr}(\mathbf{M}^T\mathbf{M})$. Since $\mathbf{J}$ and $\hat{\mathbf{J}}$ are always symmetric, to prove the theorem conclusion, we must show that $\lim_{k \to \infty} \mathbb{E}[\operatorname{tr}[\hat{\mathbf{J}}_k...\hat{\mathbf{J}}_1\hat{\mathbf{J}}_1...\hat{\mathbf{J}}_k]] = \infty$. We will achieve this by inductively proving a PSD lower bound on the expectation of this product of matrices and then leveraging the operator monotonicity of the trace. To that end, we define the matrix:

$$\mathbf{M}_r = \mathbf{J}^{2r} + \left(\frac{\eta^2}{nB} - \frac{\eta^2}{n^2}\right)^r \sum_{y_1...,y_r=1}^n \mathbf{H}_{y_r}...\mathbf{H}_{y_1}^2...\mathbf{H}_{y_r}.$$

We show that $\mathbb{E}[\hat{\mathbf{J}}_k...\hat{\mathbf{J}}_1\hat{\mathbf{J}}_1...\hat{\mathbf{J}}_k] \succeq \mathbf{M}_k \ \forall k \in \mathbb{N}$. First, let us start with the base case $r = 1$. Recall that $\mathbf{H} = \frac{1}{n}\sum_{i=1}^n \mathbf{H}_i$ and $\hat{\mathbf{H}} = \frac{1}{B}\sum_{i=1}^n x_i \mathbf{H}_i$, where $x_i \sim \text{Bern}(B/n)$.

$$
\begin{aligned}
\mathbb{E}[\hat{\mathbf{J}}_1\hat{\mathbf{J}}_1] &= \mathbb{E}[(\mathbf{I} - \eta\hat{\mathbf{H}}_1)(\mathbf{I} - \eta\hat{\mathbf{H}}_1)] \\
&= \mathbb{E}[\mathbf{I} - 2\eta\hat{\mathbf{H}}_1 + \eta^2\hat{\mathbf{H}}_1^2] \\
&= \mathbb{E}\left[\mathbf{I} - 2\left(\frac{\eta}{B}\sum_{i=1}^n x_i\mathbf{H}_i\right) + \left(\frac{\eta}{B}\sum_{i=1}^n x_i\mathbf{H}_i\right)^2\right] \\
&= \mathbf{I} - \frac{2\eta}{n}\sum_{i=1}^n \mathbf{H}_i + \frac{\eta^2}{B^2}\sum_{i,j=1}^n \mathbb{E}[x_i x_j]\mathbf{H}_i\mathbf{H}_j
\end{aligned}
$$

At this point, we have simply substituted in the relevant definitions and rearranged the terms. Now, note that $\mathbb{E}[x_i x_j] = \frac{B^2}{n^2}$ if $i \neq j$ and $\mathbb{E}[x_i x_j] = \frac{B}{n}$ otherwise. We substitute this in, and use the fact that $\mathbf{H}^2 = \left(\frac{1}{n}\sum_{i=1}^n \mathbf{H}_i\right)^2$.

$$
\begin{aligned}
\mathbb{E}[\hat{\mathbf{J}}_1\hat{\mathbf{J}}_1] &= \mathbf{I} - \frac{2\eta}{n}\sum_{i=1}^n \mathbf{H}_i + \frac{\eta^2}{nB}\sum_{i=1}^n \mathbf{H}_i^2 + \frac{\eta^2}{n^2}\sum_{i\neq j}^n \mathbf{H}_i\mathbf{H}_j \\
&= \mathbf{I} - \frac{2\eta}{n}\sum_{i=1}^n \mathbf{H}_i + \frac{\eta^2}{n^2}\sum_{i,j=1}^n \mathbf{H}_i\mathbf{H}_j + \left(\frac{\eta^2}{nB} - \frac{\eta^2}{n^2}\right)\sum_{i=1}^n \mathbf{H}_i^2 \\
&= \mathbf{I} - 2\eta\mathbf{H} + \eta^2\mathbf{H}^2 + \left(\frac{\eta^2}{nB} - \frac{\eta^2}{n^2}\right)\sum_{i=1}^n \mathbf{H}_i^2 \\
&= \mathbf{M}_1
\end{aligned}
$$

Thus, we have proved the base case $\mathbb{E}[\hat{\mathbf{J}}_1^2] \succeq \mathbf{M}_1$, which also implies $\mathbb{E}[\hat{\mathbf{J}}_k^2] \succeq \mathbf{M}_1$, since they are identically distributed. We next prove the inductive step under the assumption that $\mathbb{E}[\hat{\mathbf{J}}_{k-1}...\hat{\mathbf{J}}_1^2...\hat{\mathbf{J}}_{k-1}] \succeq \mathbf{M}_{k-1}$. By the Law of Total Expectation and linearity,

$$
\mathbb{E}[\hat{\mathbf{J}}_k...\hat{\mathbf{J}}_1^2...\hat{\mathbf{J}}_k] = \mathbb{E}[\mathbb{E}[\hat{\mathbf{J}}_k...\hat{\mathbf{J}}_1^2...\hat{\mathbf{J}}_k|\hat{\mathbf{J}}_k]] = \mathbb{E}[\hat{\mathbf{J}}_k \cdot \mathbb{E}[\hat{\mathbf{J}}_{k-1}...\hat{\mathbf{J}}_1^2...\hat{\mathbf{J}}_{k-1}] \cdot \hat{\mathbf{J}}_k] \succeq \mathbb{E}[\hat{\mathbf{J}}_k\mathbf{M}_{k-1}\hat{\mathbf{J}}_k].
$$

The arithmetic in the next part of the argument is analogous to the argument in the base case.

$$
\begin{aligned}
\mathbb{E}[\hat{\mathbf{J}}_k\mathbf{M}_{k-1}\hat{\mathbf{J}}_k] &= \mathbb{E}[(\mathbf{I} - \eta\hat{\mathbf{H}}_k)\mathbf{M}_{k-1}(\mathbf{I} - \eta\hat{\mathbf{H}}_k)] \\
&= \mathbf{M}_{k-1} - \eta\mathbf{M}_{k-1}\mathbf{H} - \eta\mathbf{H}\mathbf{M}_{k-1} + \mathbb{E}\left[\frac{\eta^2}{B^2}\sum_{i,j=1}^n x_i x_j\mathbf{H}_i\mathbf{M}_{k-1}\mathbf{H}_j\right] \\
&= \mathbf{M}_{k-1} - \eta\mathbf{M}_{k-1}\mathbf{H} - \eta\mathbf{H}\mathbf{M}_{k-1} + \frac{\eta^2}{n^2}\sum_{i,j=1}^n \mathbf{H}_i\mathbf{M}_{k-1}\mathbf{H}_j \\
&\quad + \eta^2\left(\frac{1}{nB} - \frac{1}{n^2}\right)\sum_{i=1}^n \mathbf{H}_i\mathbf{M}_{k-1}\mathbf{H}_i \\
&= \mathbf{J}_k\mathbf{M}_{k-1}\mathbf{J}_k + \eta^2\left(\frac{1}{nB} - \frac{1}{n^2}\right)\sum_{i=1}^n \mathbf{H}_i\mathbf{M}_{k-1}\mathbf{H}_i
\end{aligned}
$$

Next, we substitute in the definition of $\mathbf{M}_{k-1}$. Recall that if $\mathbf{A}$ is symmetric and $\mathbf{B}$ is PSD, then $\mathbf{A}\mathbf{B}\mathbf{A}$ is also PSD.

$$
\begin{aligned}
\mathbb{E}[\hat{\mathbf{J}}_k\mathbf{M}_{k-1}\hat{\mathbf{J}}_k] &= \mathbf{J}_k\left(\mathbf{J}^{2(k-1)} + \left(\frac{\eta^2}{nB} - \frac{\eta^2}{n^2}\right)^{k-1}\sum_{y_1...,y_{k-1}=1}^n \mathbf{H}_{y_{k-1}}...\mathbf{H}_{y_1}^2...\mathbf{H}_{y_{k-1}}\right)\mathbf{J}_k \\
&\quad + \eta^2\left(\frac{1}{nB} - \frac{1}{n^2}\right)\sum_{i=1}^n \mathbf{H}_i\left(\mathbf{J}^{2(k-1)} + \left(\frac{\eta^2}{nB} - \frac{\eta^2}{n^2}\right)^{k-1}\sum_{y_1...,y_{k-1}=1}^n \mathbf{H}_{y_{k-1}}...\mathbf{H}_{y_1}^2...\mathbf{H}_{y_{k-1}}\right)\mathbf{H}_i \\
&\succeq \mathbf{J}^{2k} + \left(\frac{\eta^2}{nB} - \frac{\eta^2}{n^2}\right)^k \sum_{y_1...,y_k=1}^n \mathbf{H}_{y_k}...\mathbf{H}_{y_1}^2...\mathbf{H}_{y_k} \\
&= \mathbf{M}_k
\end{aligned}
$$

Note that we are able to drop the cross-terms in the first step since $\mathbf{ABA} \succeq \mathbf{0}$, whenever $\mathbf{A}$ is symmetric and $\mathbf{B}$ is PSD. Therefore, we have show that $\mathbb{E}[\hat{\mathbf{J}}_k...\hat{\mathbf{J}}_1^2...\hat{\mathbf{J}}_k] \succeq \mathbf{M}_k$ for all $k \in [n]$. By the operator monotonicity and linearity of the trace,

$$\mathbb{E}[\text{tr}[\hat{\mathbf{J}}_k...\hat{\mathbf{J}}_1^2...\hat{\mathbf{J}}_k]] = \text{tr}[\mathbb{E}[\hat{\mathbf{J}}_k...\hat{\mathbf{J}}_1^2...\hat{\mathbf{J}}_k]] \geq \text{tr}[\mathbf{M}_k]$$

$$= \text{tr}[\mathbf{J}^{2k}] + \left(\frac{\eta^2}{nB} - \frac{\eta^2}{n^2}\right)^k \text{tr}\left[\sum_{y_1...,y_k=1}^n \mathbf{H}_{y_k}...\mathbf{H}_{y_1}^2...\mathbf{H}_{y_k}\right]$$

$$= \text{tr}[\mathbf{J}^{2k}] + \left(\frac{\eta^2}{nB} - \frac{\eta^2}{n^2}\right)^k \sum_{y_1...,y_k=1}^n \|\mathbf{H}_{y_k}...\mathbf{H}_{y_1}\|_F^2$$

Hence, we have proved part (i) of the lemma. We will now prove part (ii). Define the following matrix:

$$\mathbf{N}_r = \left(\frac{\eta^2}{nB} - \frac{\eta^2}{n^2}\right)^r \sum_{y_1...,y_r=1}^n \mathbf{H}_{y_r}...\mathbf{H}_{y_1}^2...\mathbf{H}_{y_r},$$

and define $\mathbf{N}_0 = \mathbf{I}$. We will prove that:

$$\mathbb{E}[\hat{\mathbf{J}}_k...\hat{\mathbf{J}}_1^2...\hat{\mathbf{J}}_k] \preceq \sum_{r=0}^k (1-\epsilon)^{2(k-r)} \binom{k}{r} \mathbf{N}_r, \tag{3}$$

for all $k \in \mathbb{N}$, where $\epsilon > 0$. First, note that the condition of the theorem implies $-(1-\epsilon)\mathbf{I} \prec \mathbf{J} \prec (1-\epsilon)\mathbf{I}$ for some $\epsilon > 0$. First, note that the base case:

$$\mathbb{E}[\hat{\mathbf{J}}_1^2] \preceq (1-\epsilon)^2 \mathbf{N}_0 + \mathbf{N}_1 = (1-\epsilon)^2 \mathbf{I} + \left(\frac{\eta^2}{nB} - \frac{\eta^2}{n^2}\right) \sum_{i=1}^n \mathbf{H}_i^2,$$

holds from our computations in part (i), since $\mathbf{J}^2 \preceq (1-\epsilon)^2 \mathbf{N}_0$. Now for the inductive step. Under the assumption that Equation 3 holds for $k-1$, we show it also hold for $k$. By the same argument as in part (i), we find that:

$$\mathbb{E}[\hat{\mathbf{J}}_k...\hat{\mathbf{J}}_1^2...\hat{\mathbf{J}}_k] \preceq \mathbb{E}\left[\hat{\mathbf{J}}_k \left(\sum_{r=0}^{k-1}(1-\epsilon)^{2(k-1-r)} \binom{k-1}{r} \mathbf{N}_r\right) \hat{\mathbf{J}}_k\right]$$

$$= \mathbf{J}_k \left(\sum_{r=0}^{k-1}(1-\epsilon)^{2(k-1-r)} \binom{k-1}{r} \mathbf{N}_r\right) \mathbf{J}_k$$

$$+ \left(\frac{\eta^2}{nB} - \frac{\eta^2}{n^2}\right) \sum_{i=1}^n \mathbf{H}_i \left(\sum_{r=0}^{k-1}(1-\epsilon)^{2(k-1-r)} \binom{k-1}{r} \mathbf{N}_r\right) \mathbf{H}_i$$

$$\preceq (1-\epsilon)^2 \left(\sum_{r=0}^{k-1}(1-\epsilon)^{2(k-1-r)} \binom{k-1}{r} \mathbf{N}_r\right)$$

$$+ \left(\frac{\eta^2}{nB} - \frac{\eta^2}{n^2}\right) \sum_{i=1}^n \mathbf{H}_i \left(\sum_{r=0}^{k-1}(1-\epsilon)^{2(k-1-r)} \binom{k-1}{r} \mathbf{N}_r\right) \mathbf{H}_i$$

$$= \sum_{r=0}^{k-1}(1-\epsilon)^{2(k-r)} \binom{k-1}{r} \mathbf{N}_r$$

$$+ \sum_{r=0}^{k-1}(1-\epsilon)^{2(k-1-r)} \binom{k-1}{r} \cdot \left(\frac{\eta^2}{nB} - \frac{\eta^2}{n^2}\right) \sum_{i=1}^n \mathbf{H}_i \mathbf{N}_r \mathbf{H}_i$$

$$= \sum_{r=0}^{k-1}(1-\epsilon)^{2(k-r)} \binom{k-1}{r} \mathbf{N}_r + \sum_{r=0}^{k-1}(1-\epsilon)^{2(k-1-r)} \binom{k-1}{r} \mathbf{N}_{r+1},$$

where the last line follows from $\left(\frac{\eta^2}{nB} - \frac{\eta^2}{n^2}\right) \sum_{i=1}^n \mathbf{H}_i \mathbf{N}_r \mathbf{H}_i = \mathbf{N}_{r+1}$. We can rewrite the second summation to go from $r = 1$ to $r = k$ by replacing $r + 1$ with $r$ to combine the terms like so:

$$\sum_{r=0}^{k-1} (1-\epsilon)^{2(k-r)} \binom{k-1}{r} \mathbf{N}_r + \sum_{r=1}^{k} (1-\epsilon)^{2(k-r)} \binom{k-1}{r-1} \mathbf{N}_r$$

$$= (1-\epsilon)^{2k} \mathbf{N}_0 + \sum_{r=1}^{k-1} \left[ (1-\epsilon)^{2(k-r)} \binom{k-1}{r} + (1-\epsilon)^{2(k-r)} \binom{k-1}{r-1} \right] \mathbf{N}_r + \mathbf{N}_k$$

$$= (1-\epsilon)^{2k} \mathbf{N}_0 + \sum_{r=1}^{k-1} (1-\epsilon)^{2(k-r)} \binom{k}{r} \mathbf{N}_r + \mathbf{N}_k$$

$$= \sum_{r=0}^{k} (1-\epsilon)^{2(k-r)} \binom{k}{r} \mathbf{N}_r.$$

Hence, we have completed the proof by induction. Note that to go from the second to the third line in the previous equation block, we applied the recursive formula for binomial coefficients (see Fact 2). Now, doing the same as before, we can conclude that:

$$\mathbb{E} \, \mathrm{tr}[\hat{\mathbf{J}}_k ... \hat{\mathbf{J}}_1^2 ... \hat{\mathbf{J}}_k] \leq \sum_{r=0}^{k} (1-\epsilon)^{2(k-r)} \binom{k}{r} \mathrm{tr}[\mathbf{N}_r].$$

By the condition of the theorem that $\lim_{k\to\infty} \left(\frac{\eta^2}{nB} - \frac{\eta^2}{n^2}\right)^k \sum_{y_1...,y_k=1}^n \|\mathbf{H}_{y_k}...\mathbf{H}_{y_1}\|_F^2 = 0$, we have that:

$$\mathrm{tr}[\mathbf{N}_r] = \left(\frac{\eta^2}{nB} - \frac{\eta^2}{n^2}\right)^r \sum_{y_1...,y_r=1}^n \mathrm{tr}[\mathbf{H}_{y_r}...\mathbf{H}_{y_1}^2...\mathbf{H}_{y_r}]$$

$$= \left(\frac{\eta^2}{nB} - \frac{\eta^2}{n^2}\right)^r \sum_{y_1...,y_r=1}^n \|\mathbf{H}_{y_r}...\mathbf{H}_{y_1}\|_F^2,$$

goes to zero as $r \to \infty$. Let $n_r = \mathrm{tr}[\mathbf{N}_r]$ and $\delta = (1-\epsilon)^2$. To prove the theorem statement, it suffices to show that $\lim_{k\to\infty} \sum_{r=0}^{k} \delta^{k-r} \binom{k}{r} n_r = 0$, when $\delta \in (0, 1)$ and $(n_r)_{r\in\mathbb{N}}$ is a sequence of positive real numbers converging to zero. By the theorem statement, we know that $\frac{1}{\epsilon^r} \mathrm{tr}[\mathbf{N}_r] \to 0$. Therefore, there exists some constant $C$ such that $\frac{1}{\epsilon^r} \mathrm{tr}[\mathbf{N}_r] \leq C$ for all $r \in [n]$, which implies $\mathrm{tr}[\mathbf{N}_r] \leq \epsilon^r C$. Therefore, by the Binomial formula:

$$\sum_{r=0}^{k} \delta^{k-r} \binom{k}{r} n_r \leq \sum_{r=0}^{k} \binom{k}{r} (1-\epsilon^2)^{k-r} \cdot C\epsilon^r = C \left((1-\epsilon)^2 + \epsilon\right)^k.$$

This term must go to zero as $k \to \infty$, since $(1-\epsilon)^2 + \epsilon = 1 - \epsilon + \epsilon^2 < 1$. $\qquad\square$

## B    RESULT OF WU ET AL. (2022)

Here, we summarize the relevant information needed to state Theorem 3.3 of Wu et al. (2022), which considers the settings where one has access to a dataset of the form $(\mathbf{x}_i, y_i)_{i\in[n]}$, where $\mathbf{x}_i \in \mathbb{R}^m$, $y_i \in \mathbb{R}$. $f(\mathbf{x}_i, \mathbf{w})$ is a function parameterized by $\mathbf{w}$ such that $f : \mathbb{R}^m \times \mathbb{R}^d \to \mathbb{R}$. The loss function applied to each sample is MSE, denoted by $\ell_i(\mathbf{w}) = |f(\mathbf{x}_i, \mathbf{w}) - y_i|^2$, and the total loss over all the samples is given by $L(\mathbf{w}) = \frac{1}{n} \sum_{i=1}^n \ell_i(\mathbf{w})$. The linearized SGD dynamics considered is the same as Definition 1, except that $\mathcal{S} \subset [n]$ is sampled from all size $B$ subsets of $[n]$ uniformly.

One can now define two critical matrices. Let $\mathbf{\Sigma}(\mathbf{w}) = \frac{1}{n} \sum_{i=1}^n \nabla \ell_i(\mathbf{w}) \nabla \ell_i(\mathbf{w})^T - \nabla L(\mathbf{w}) \nabla L(\mathbf{w})^T$ be the noise covariance matrix, and let $\mathbf{G}(\mathbf{w}) = \frac{1}{n} \sum_{i=1}^n \nabla f(\mathbf{x}_i, \mathbf{w}) \nabla f(\mathbf{x}_i, \mathbf{w})^T$ be the Fisher matrix that characterizes the local geometry of the loss landscape. Next, Wu et al. (2022) define the loss-scaled alignment factor (Equation 3 in Wu et al. (2022)):

$$\mu(\mathbf{w}) = \frac{\mathrm{tr}[\mathbf{\Sigma}(\mathbf{w})\mathbf{G}(\mathbf{w})]}{2L(\mathbf{w})\|\mathbf{G}(\mathbf{w})\|_F^2}. \tag{4}$$

The last step before stating the theorem is the introduction of the definition of linear stability considered in Wu et al. (2022).

**Definition 4.** *(Linear stability in Wu et al. (2022)) A global minimum $\mathbf{w}^*$ is said to be linearly stable if there exists a $C > 0$ such that it holds for linearized dynamics that $\mathbb{E}[L(\mathbf{w}_t)] \leq C \cdot \mathbb{E}[L(\mathbf{w}_0)] \ \forall t \geq 0$, with $\mathbf{w}_0$ being sufficiently close to $\mathbf{w}^*$.*

Finally, the theorem follows.

**Theorem 3.** *(Theorem 3.3 in Wu et al. (2022)) Let $\mathbf{w}^*$ be a global minima that is linearly stable (Definition 4). Denote by $\mu(\mathbf{w})$ the alignment factors for linearized SGD (Equation 4). If $\mu(\mathbf{w}) > \mu_0$, then $\|\mathbf{H}(\mathbf{w}^*)\|_F \leq \frac{1}{\eta}\sqrt{\frac{B}{\mu_0}}$.*

