# OpenReview forum: "A Precise Characterization of SGD Stability Using Loss Surface Geometry"
_ICLR.cc/2024/Conference — ICLR 2024 poster_

### Official Review · Reviewer_SPXN · 2023-10-31

**Soundness:** 3 good
**Presentation:** 2 fair
**Contribution:** 3 good
**Rating:** 6
**Confidence:** 4

**Summary:**

This manuscript investigates the linear stability of SGD and obtain a sufficient condition for instability (equivalently, a necessary condition for stability).
The authors introduce a coherence measure $\sigma$ to measure the strength of  alignment among Hessian matrices. Using this measure, they derive their main result Theorem 1, which, as they claim, is more general than Theorem 3.3 in [Wu et al, 2022]. The authors also show that Theorem 1 is nearly optimal given that $\sigma$ and $n/B$ are $O(1)$ quantities. Some experiments are carried out to support the theoretical results.

**Strengths:**

On a whole I think these results are neat, novel and have their own advantages.  The characterization seems to be rather precise and Lemma 4.1 is of particular interest.

**Weaknesses:**

- The writing of this manuscript needs to be improved. In particular, the many details are present in an unclear way and I find it very hard to follow them smoothly. The citations and references also need to be re-organized. See Question section for more details.

- The definition of ``Coherence measure'' is not intuitive.  Personally, I do not understand why the proposed definition can quantify the coherence among Hessian matrices. In particular, the authors claim that $\lambda_1(H_i)$ is the i-th diagonal entry of $S$. This is obviously not true unless that $H_i$ is rank-1.

- The authors might make a better interpretation of Theorem 1. What does this result imply about the implicit regularization of SGD (beyond that of GD)? How stability is related to the hyperparameters, e.g. $\eta$, $B$, and alignment of $H_i$? Some relevant discussion can be found in the experiment part Section 5.2, but I think it would be better to provide some intuition right after Theorem 1.

 - In Section 3.2.1 the authors compare Theorem 1 to Theorem 3.3 in [Wu et al, 2022], and stated the advantages of their result. Among these stated advantages,
   - The first point makes sense to me.
   - In the second point, why do you say ``This definition is notably weaker than our notion of stability''?
   - By the third point, you seem to imply that the bound in Theorem 1 is sharper in the low-rank case. But what is the point in considering $\sigma$ equal to one? To me, the third point is an unclear comparison between two results, which cannot prove the advantage of Theorem 1.


- Theorem 2, the optimality of Theorem 1, strongly relies on the condition that $\sigma, n/B = O(1)$. There are two concerns:
   - In Theorem 2 the authors assume that $\sigma \in [n]$. Is there anything to guarantee $\sigma \leq n$? I don't think it is clear from the definition of $\sigma$. Is $\sigma$ inherently bounded? Moreover, do you mean $\sigma\leq n$? Hence, it is unclear what the assumption $\sigma=O(1)$ means.
   - Also, I do not think it is natural to assume $n/B=O(1)$ as usually $B\ll n$.

**Questions:**

- In paragraph 1, when introducing the concept of "implicit bias", instead of citing (Li et al., 2022), I think it is more appropriate to cite the seminar works  (Neyshabur et al., arXiv:1412.6614) and (Zhang et al., ICLR 2017).
 - In paragraph 2, when citing empirical works on relating sharpness to generalization, I think the important comprehensive investigations by (Jiang et al., ICLR2020) is missed.
 - In paragraph 4, when stating "GD ... empirically validated to predict the sharpness of overparameterized neural network", the author cites (Cohen et al., 2021). However, this empirical predictability of linear stability analysis has been observed in (Wu et al., NeurIPS2018).
 - In Section 2
 	- In paragraph 1, when stating the rationale for assuming over-parameterization, the authors cite the work (Allen-Zhu et al., 2019). This seems quite strange to me.
 	- In Definition 1,  it is unclear whether the sampling is done with or without replacement.
- In Section 3
	- What is the $\frac{1}{B}\sum_{i=1}^n x_i H_i$ in the second paragraph.
	- In Theorem 1, what does the subscript in $\hat{J}_i$ stand for? Complexity measure => coherence measure.
	- In Theorem 2, what do you mean $\sigma\in [n]$? Is $\sigma$ a real value?

---

> ### Author Response · Authors · 2023-11-16
>
> We thank the reviewer for their insightful comments.
>
> > The writing of this manuscript needs to be improved. In particular, the many details are present in an unclear way and I find it very hard to follow them smoothly. The citations and references also need to be re-organized. See Question section for more details.
>
> Below we have responded to your points in the Question section.
>
> > The definition of ``Coherence measure'' is not intuitive. Personally, I do not understand why the proposed definition can quantify the coherence among Hessian matrices. In particular, the authors claim that $\lambda_1(H_i)$ is the i-th diagonal entry of $S$. This is obviously not true unless that $H_i$ is rank-1.
>
> Thank you for pointing out this omission. We have added that the $\{H_i\}$ are rank-1 in this example and additional explanation for why it is reasonable to consider this case.
>
> For binary classification with $f$ denoting a scalar scoring function of weights and loss being an $\mathbb{R} \rightarrow \mathbb{R}$ function of $f$, the Hessian is the sum of a rank-one term and a full second order term. However, in over-parameterized systems, each $f_i$ attains its least value and hence the full second order term is zero, and so $H_i$ being rank-one is a good assumption. (See eqn. (13) for additional details here: https://www.cs.toronto.edu/~rgrosse/courses/csc2541_2021/readings/L02_Taylor_approximations.pdf)
>
> With this in mind, we refer to Section 3.1 of our paper which explains the intuition of the coherence measure using two edge cases where the $H_i$ are rank-1.
>
> > The authors might make a better interpretation of Theorem 1. What does this result imply about the implicit regularization of SGD (beyond that of GD)? How stability is related to the hyperparameters, e.g. $\eta$, $B$, and alignment of $H_i$? Some relevant discussion can be found in the experiment part Section 5.2, but I think it would be better to provide some intuition right after Theorem 1.
>
> We agree it is best to immediately explain the key takeaways after the theorem statement.
>
> We discuss how our analysis implies a "squared scaling rule" between $B$ and $\eta$ after Lemma 4.1, but we now also mention this after Theorem 1. Another point to note regarding how SGD is regularized beyond GD is that GD does not depend on the coherence measure $\sigma$. We give examples in the paragraph above Theorem 1 demonstrating the importance of the Hessian alignment (captured by $\sigma$) in characterizing SGD stability. We now emphasize this more explicitly after the theorem statement.
>
> > In Section 3.2.1 the authors compare Theorem 1 to Theorem 3.3 in [Wu et al, 2022], and stated the advantages of their result. Among these stated advantages, 1) The first point makes sense to me. 2) In the second point, why do you say ``This definition is notably weaker than our notion of stability''? 3) By the third point, you seem to imply that the bound in Theorem 1 is sharper in the low-rank case. But what is the point in considering $\sigma$ equal to one? To me, the third point is an unclear comparison between two results, which cannot prove the advantage of Theorem 1.
>
> Regarding (2), the notion of stability Wu et. al. uses (and restated in Appendix B of our paper) is that $\mathbb{E}[L(w_t)] \leq C E[L(w_0)]$ for some constant $C$ and for all $t \geq 0$. This is weaker than the typical expected mean-squared sense of stability, since one can choose $L(\cdot)$ such that this definition is satisfied but $\||w_t - w^*\||_2 \rightarrow \infty$.
>
> We chose $\sigma,\mu = 1$ for simplicity, but the important takeaway from this point is that when these quantities are held constant and stable rank is bounded, then our condition becomes more general than that of [Wu et. al., 2022] as $n$ increases. While you are correct this does not prove a general advantage, it does prove an advantage in an asymptotic sense for these fixed parameter problems, which we would  argue is a theoretically interesting regime.

---

> ### Author Response · Authors · 2023-11-16
>
> > Theorem 2, the optimality of Theorem 1, strongly relies on the condition that $\sigma, B/n = O(1)$. There are two concerns: 1) In Theorem 2 the authors assume that $\sigma \in [n]$. Is there anything to guarantee $\sigma \leq n$? I don't think it is clear from the definition of $\sigma$. Is $\sigma$ inherently bounded? Moreover, do you mean $\sigma \leq n$? Hence, it is unclear what the assumption $\sigma = O(1)$ means. 2) Also, I do not think it is natural to assume $B/n = O(1)$ as usually $B \ll n$.
>
> Theorem 2 is provided to give an idea of how sharp our current analysis is and what approaches would be needed to improve it. To clarify, we do not assume that $\sigma$ is a specific value, rather we state that there exists an input problem with Hessians $\\{H\_i\\}\_{i \in [n]}$ such that $\sigma$ equals this value and the linearized dynamics don't diverge. This means any improvement to Theorem 1 cannot imply that these problems diverge.
>
> The discussion above Theorem 2 where we consider $\sigma, n/B = O(1)$ is just pointing out that the conditions of Theorem 1 and Theorem 2 are close in this parameter range, and so we cannot expect much improvement in this regime. However, Theorem 2 does not make any assumption regarding this. From a theoretical perspective, we think it is natural to consider this regime to understand its behavior. However, we agree that the case where $B \ll n$ is important to understand in future work, and so we have added a sentence pointing this out above Theorem 2.
>
> > In paragraph 1, when introducing the concept of "implicit bias", instead of citing (Li et al., 2022), I think it is more appropriate to cite the seminar works (Neyshabur et al., arXiv:1412.6614) and (Zhang et al., ICLR 2017).
>
> Thank you, we have included these references in the updated manuscript.
>
> > In paragraph 2, when citing empirical works on relating sharpness to generalization, I think the important comprehensive investigations by (Jiang et al., ICLR2020) is missed.
>
> Of course, this is a seminal paper that is worth referring to. Thank you for pointing this out.
>
> > In paragraph 4, when stating "GD ... empirically validated to predict the sharpness of overparameterized neural network", the author cites (Cohen et al., 2021). However, this empirical predictability of linear stability analysis has been observed in (Wu et al., NeurIPS2018).
>
> We apologize for missing this important reference in this place. We have referred to the same paper elsewhere in other contexts.
>
> > In paragraph 1, when stating the rationale for assuming over-parameterization, the authors cite the work (Allen-Zhu et al., 2019). This seems quite strange to me.
>
> We are happy to add additional references, e.g. Zhang et al 2016.
>
> > In Definition 1, it is unclear whether the sampling is done with or without replacement.
>
> We are using Bernoulli sampling which is not the same as sampling with or without replacement. The text under eqn. (1) in Definition 1 describes the probabilistic sampling model we work with. To clarify $\mathcal{S}$ is a set (so there are no repetitions) and $i \in \mathcal{S}$ being independent from $j \in \mathcal{S}$ implies it cannot be sampling without replacement.
>
> > What is the $\frac{1}{B}\sum_{i=1}^n x_i H_i$ in the second paragraph.
>
> Thanks for pointing this typo out. We have changed it to $\frac{1}{B} \sum_{i \in \mathcal{S}} H_i$. We reformulate this later in the proof as $\frac{1}{B}\sum_{i=1}^n x_i H_i$ where the $x_i$ are Bernoulli random variables, but you are correct we have not defined it at this point.
>
> > In Theorem 1, what does the subscript in $\hat{J}_i$ stand for?
>
> We have changed "Let $\hat{J}\_i$ be independent Jacobians of SGD dynamics" to "Let $\\{\hat{J}\_i\\}\_{i\in\mathbb{N}}$ be a sequence of i.i.d. copies of $\hat{J}$ defined in Definition 1" to make it clearer we consider an infinite sequence of i.i.d. samples of the stochastic Jacobian.
>
> > In Theorem 2, what do you mean $\sigma \in [n]$? Is $\sigma$ a real value?
>
> For an arbitrary set of Hessians $\{H_i\}_{i\in [n]}$, the coherence measure $\sigma$ can be a general real value. However, Theorem 2 is an existential result that says for any chosen $\sigma \in \{1,2,...,n\}$, there exists a set of Hessians $\{H_i\}$ satisfying the properties described in the theorem.

---

### Official Review · Reviewer_ELtf · 2023-10-31

**Soundness:** 4 excellent
**Presentation:** 4 excellent
**Contribution:** 4 excellent
**Rating:** 8
**Confidence:** 4

**Summary:**

The paper analyzes the linear stability of SGD of any additively decomposable loss function around the minimum $w^*$. The paper then derives a necessary and sufficient condition for the (in)stability, which relies on a novel *coherence measure* $\sigma$, which is, in turn, intuitively connected to the alignment property of the collection of Hessians (of individual loss function). This is verified experimentally.

**Strengths:**

1. Very well-written. I especially liked how the theoretical discussions are backed with intuitions and specific examples. I very much enjoyed reading this work.
2. Although the mathematics used here are rather elementary, the theoretical discussions are complete; a necessary-sufficient condition for stability is provided, and the motivation and intuition behind the coherence measure $\sigma$ are well-explained. (Elementaryness is a plus for me)
3. Clear contribution and a good advancement in the linear stability analysis of SGD

**Weaknesses:**

1. The analysis relies on the Bernoulli sampling model, which, although is in expectation the same as with replacement sampling or uniformly sampling from all $B$-sets, still is a bit different as the size of $\mathcal{S}$ itself now becomes random. Have the authors tried to consider multinomial distribution as the sampling distribution?
2. Moreover, without replacement sampling where the event $i \in \mathcal{S}$ is dependent on $j \in \mathcal{S}$ (depending on the order), there should be some theoretical discussions on the effect of using these two (most-widely-used) random sampling schemes. (I appreciate that the paper has experiments on this, but then, theorem-wise, what would possibly change?)

**Questions:**

1. The analyses presented here are solely on the stability of the iterates, i.e., whether they diverge or not. Is there any chance that this gives some insight into whether they converge? Even further, depending on the alignment of the Hessians, can we say something about the local convergence rate?
2. The relationship between batch size and learning rate that I'm more familiar with (e.g., starting from Goyal et al. (2017)) is the linear scaling rule, but here it is shown to be squared, which has also been reported in the past (e.g., Krizhevsky (2014)). Can the authors elaborate on why this stability analysis leads to squared? Then, at least locally, is squared scaling law the way to go, given that the Taylor expansion is accurate?
3. In Figure 1, for small batch sizes, there is a large gap between Theorem 2 and the red boundary for $\eta = 0.8$. Any particular reason for this?
4. How would this be extended to momentum gradient descent or any of the adaptive gradient methods (e.g., Adam)? If the time allows, can the authors provide experiments for these as well in the same setting?
5. In the prior works on linear stability, were there any quantities that resemble $\sigma$ in their role? If there were, can the authors compare those to the proposed $\sigma$?

---

> ### Author Response · Authors · 2023-11-14
>
> We thank the reviewer for their insightful comments.
>
> > The analysis relies on the Bernoulli sampling model, which, although is in expectation the same as with replacement sampling or uniformly sampling from all $B$-sets, still is a bit different as the size of $\mathcal{S}$ itself now becomes random. Have the authors tried to consider multinomial distribution as the sampling distribution?
>
> > Moreover, $i \in \mathcal{S}$ without replacement sampling where the event is dependent on $j \in \mathcal{S}$ (depending on the order), there should be some theoretical discussions on the effect of using these two (most-widely-used) random sampling schemes. (I appreciate that the paper has experiments on this, but then, theorem-wise, what would possibly change?)
>
> We are optimistic that improved understanding of the Bernoulli sampling model can provide useful insight to other sampling models, as was the case for the previous line of work on matrix completion by Candes, Tao, and others which culminated in Benjamin Recht's seminal paper [Recht, 2011]. We also note that sampling with/without replacement are also just theoretical approximations of the typical true sampling behavior, where the data is shuffled and iterated over in an epoch. In reality, the sampling is often not even independent between iterations.
>
> Lemma 4.1 could likely be rewritten using the sampling without replacement formulation for $\hat{J}$ used in [Ma and Ying, 2021]. However, there doesn't seem to be an obvious way to simplify the expressions, and we would be stuck with a characterization that is no more interpretable than [Ma and Ying, 2021].
>
> Another possibility would be to rewrite Lemma 4.1 for sampling with replacement. We expect that the coefficients depending on $\eta$ and $B$ would change, but the proof would generally stay the same. That is, let $x_1,...,x_B$ each be uniformly sampled from $[n]$. Then, $\mathbb{E}[H_{x_i}H_{x_j}] = \frac{1}{n}\sum_k {H_k^2}$ when $i = j$ and $\mathbb{E}[H_{x_i}H_{x_j}] = \frac{1}{n^2}H^2$ when $i \neq j$. Therefore, the proof of Lemma 4.1 would start the same way when $\hat{H}=\sum_{i=1}^B H_{x_i}$. Results for the sampling with replacement case would immediately imply asymptotic results for sampling without replacement, since the total variation distance between the two distributions goes to zero as $n \rightarrow \infty$ and $B/n$ converges to a constant.
>
>
> > The analyses presented here are solely on the stability of the iterates, i.e., whether they diverge or not. Is there any chance that this gives some insight into whether they converge? Even further, depending on the alignment of the Hessians, can we say something about the local convergence rate?
>
> Lemma 4.1 provides a sufficient condition to guarantee the linearized dynamics converge $w^*$. As we simplified part (i) of the Lemma using the Hessian alignment to prove Theorem 1, it is likely that similar manipulations could be used to provide sufficient conditions for convergence of the linear dynamics depending on the Hessian alignment.
>
> We did not pursue this direction further since there seems to be less interest in this convergence behavior in prior work on linear stability of SGD. However, relating our analysis to local convergence rates of SGD is an interesting idea. For quadratic functions, this should follow directly. Extending this further would be a very interesting future direction, so we have added it to our conclusion as an open direction.
>
> > The relationship between batch size and learning rate that I'm more familiar with (e.g., starting from Goyal et al. (2017)) is the linear scaling rule, but here it is shown to be squared, which has also been reported in the past (e.g., Krizhevsky (2014)). Can the authors elaborate on why this stability analysis leads to squared? Then, at least locally, is squared scaling law the way to go, given that the Taylor expansion is accurate?
>
> The derivation of $\mathbb{E}[\hat{J}^2]$ at the beginning of the proof for Lemma 4.1 may give an idea of how the squared scaling happens, particularly by considering the case where $H_i$ are $1 \times 1$. This emphasizes a connection to the additivity versus scalar multiplication in the variance of a random variable, i.e., $\operatorname{Var}(X+Y) = \operatorname{Var}(X) + \operatorname{Var}(Y)$ and $\operatorname{Var}(cX) = c^2\operatorname{Var}(X)$, where $X,Y$ are independent random variables and $c \in \mathbb{R}$ is a constant. We can interpret $\eta$ as a scalar multiplier in a sum of random variables and $B$ as the number of random variables being summed. This seems to imply that when the Taylor series truncation and Bernoulli dynamics are good approximations, we expect squared scaling to hold.

---

> ### Author Response · Authors · 2023-11-14
>
> > In Figure 1, for small batch sizes, there is a large gap between Theorem 2 and the red boundary for $\eta = 0.8$. Any particular reason for this?
>
> We conjecture that this may be due to the discrepancy between Bernoulli sampling used in the theorem and sampling without replacement used in the experiments. Specifically when $B$ is small, differences between the two sampling methods may have more of an effect due to the higher relative variance in the effective batch size. In the case where $\eta=0.8$ is near the limit of GD instability, the stability condition seems more sensitive to this difference in effective batch size.
>
> > How would this be extended to momentum gradient descent or any of the adaptive gradient methods (e.g., Adam)? If the time allows, can the authors provide experiments for these as well in the same setting?
>
> Unfortunately, our experiments are coded in a way to run SGD on the described quadratic functions using minimal space/time, and as such, we cannot easily change the optimization method during the discussion period. However, we have added this idea as an open direction in our conclusion, as we agree it would be interesting to understanding how momentum affects the stability of SGD.
>
> > In the prior works on linear stability, were there any quantities that resemble
> $\sigma$ in their role? If there were, can the authors compare those to the proposed $\sigma$?
>
> The alignment factor $\mu$ of [Wu et. al., 2022] is a measure that is similar in spirit to our $\sigma$ measure. We have described this measure in Appendix B of our paper (see eqn. (4)).
>
> While this measure is also used to quantify the stability of SGD, its mathematical formulation is significantly different from ours. It is specific to MSE loss, and it measures the alignment of the gradients for the regression output and the empirical covariance of the MSE loss gradients. Meanwhile, our measure is defined through the Hessians of an additively decomposable loss function, and it applies to this general class of functions. It is difficult to push the comparison much beyond this.

---

> > ### Comment · Reviewer_ELtf · 2023-11-23
> >
> > I thank the authors for answering my concerns. I'm satisfied with the answers and inclined to keep my score.

---

### Official Review · Reviewer_mpeb · 2023-11-01

**Soundness:** 2 fair
**Presentation:** 1 poor
**Contribution:** 1 poor
**Rating:** 3
**Confidence:** 4

**Summary:**

In this paper, the authors considered quadratic approximations of the loss function around the zero-loss manifold that allows the definition of the 'linearized SGD dynamics' (given in Equation 1). This quadratic approximation allows the definition of the coherence measure (Definition 1), which is used in the statement of Theorem 1: SGD dynamics diverge when the coherence measure lower bounds the first eigenvalue of the Hessian of the loss function. In Theorem 2, they provide a partial converse to their proven divergence results in Theorem 1. The paper concludes with some experiments.

**Strengths:**

- an important problem on the stability of SGD
- having experiments

**Weaknesses:**

The introduction section (i.e., the first two pages) is poorly written. For example, the definition of 'linearized dynamics of SGD' is missing and is just referred to in Section 2. But this is probably one of the most necessary things one needs to know to follow the paper. Moreover, the section 'contributions' is also vague. Instead of long sentences, it's better to use a concise way to deliver the message. It's fairly impossible to identify the contributions of the paper based on that section (before reading the whole paper).


Section 2: the approximation is called 'linearized dynamics,' but I think this is not the right word since you are essentially approximating the loss function with a quadratic function. Moreover, this dynamics only happens if you project to the zero-loss manifold at each step; otherwise, there is a term relating to the gradient. As a result, the setting is quite limited to only quadratic loss functions.


The word 'stability of dynamics' is used frequently in the paper while not being explained in the early pages.

**Questions:**

- the font used in the paper looks weird; I think the authors have changed something in the latex code

- In Theorem 1, $\hat{J}_i$ is used, while it is never defined (the reference to Definition 1 only defines $\hat{J}$).

- After Theorem 1, why the expectation and the arg max are equal? Also, how does divergence allow you to conclude that for almost all initial parameters, SGD convergences? These claims are not clear and are vague.





------------------------------------------------------------------------------------------------------------


After the rebuttal: I appreciate the authors for their response. They partially answered some of my concerns but this paper is still not well-written, in my opinion. The authors only referred me to another reviewer for this part of my comments which I think is not an approrpiate response.

---

> ### Author Response · Authors · 2023-11-14
>
> We thank the reviewer for their insightful comments.
>
> > The introduction section (i.e., the first two pages) is poorly written. For example, the definition of 'linearized dynamics of SGD' is missing and is just referred to in Section 2. But this is probably one of the most necessary things one needs to know to follow the paper.
>
> We believe it is more appropriate to reference Section 2 for the formal definition of the linearized dynamics, as we have done, rather than interrupt the introduction. Our paper advances a long line of work on characterizing the linear stability of SGD. As such, we believe readers who would be most interested in our paper likely have some familiarity with linear stability analysis. We believe our choice in this tradeoff is worthwhile, and Reviewer ELtf finds our paper to be "very well-written".
>
> > Moreover, the section 'contributions' is also vague. Instead of long sentences, it's better to use a concise way to deliver the message. It's fairly impossible to identify the contributions of the paper based on that section (before reading the whole paper).
>
> Our contributions give a high-level description of our contributions that separates our results from prior work in the area. We have made an effort to describe our contributions as concisely as possible while including all important information by leaving the technical details to later sections. We would be happy to make changes if you have any specific recommendations for how the contributions could be rephrased.
>
> > Section 2: the approximation is called `linearized dynamics,' but I think this is not the right word since you are essentially approximating the loss function with a quadratic function.
>
> The term "linearized dynamics" is a standard term in the study of dynamical systems; we refer to the book ``Discrete Dynamical Systems'' by Oded Galor as an introductory text to this area. It is called this because the dynamics, i.e., $w_{t+1} = \hat{J}w_t$, is a linear function from $\mathbb{R}^n$ to $\mathbb{R}^n$. The linearized dynamics comes by direct first order Taylor series of the dynamics around a stationary point; we mentioned the quadratic loss approximation route only because it could help understanding through the loss. This term has also been used by many previous works analyzing SGD including [Wu, 2022], [Jastrzebski, 2019], [Ma and Ying, 2021], etc.
>
>
> > Moreover, this dynamics only happens if you project to the zero-loss manifold at each step; otherwise, there is a term relating to the gradient. As a result, the setting is quite limited to only quadratic loss functions.
>
> This is not correct. The gradient is represented by the linear approximation $g\approx Hw$. This is a fundamental technique used in the analysis of dynamical systems. We have also cited numerous other papers in the intro that uses this approximation specifically in the context of analyzing general SGD dynamics, e.g., [Wu, 2022], [Jastrzebski, 2019], [Ma and Ying, 2021], etc.
>
> > The word 'stability of dynamics' is used frequently in the paper while not being explained in the early pages.
>
> In the paragraph before our contributions section, we have added the sentence: "We consider mean-squared stability, that is, $w^*$ is considered unstable if iterates of SGD diverge from $w^*$ under the $\ell_2$-norm in expectation".
>
> > the font used in the paper looks weird; I think the authors have changed something in the latex code
>
> We have double-checked our paper to ensure it follows all ICLR formatting guidelines and made any necessary changes.
>
> > In Theorem 1, $\hat{J}_i$ is used, while it is never defined (the reference to Definition 1 only defines $\hat{J}$).
>
> We have changed "Let $\hat{J}_i$ be independent Jacobians of SGD dynamics" to "Let $\\{\hat{J}\_i\\}\_{i\in\mathbb{N}} $ be a sequence of i.i.d. copies of $\hat{J}$ defined in Definition 1" to make it clearer we consider an infinite sequence of i.i.d. samples of the stochastic Jacobian.
>
> > After Theorem 1, why the expectation and the arg max are equal?
>
> We have changed the "argmax" to "max". For any matrix $M \in \mathbb{R}^{n \times n}$, $\||M\||_2 = \max_w \||Mw\||_2$ such that $\||w\||_2 = 1$ by definition of the $\ell_2$-induced vector norm. Hence, our statement immediately follows from our definition of the linearized dynamics, i.e., $w_t = \hat{J}_t....\hat{J}_1w_0$.
>
> > Also, how does divergence allow you to conclude that for almost all initial parameters, SGD convergences? These claims are not clear and are vague.
>
> If you are referring to our statement that $\||w_t\||_2 \rightarrow \infty$ for almost all starting $w_0$ points under the conditions of Theorem 1, this follows due to the fact that the set of vectors orthogonal to the maximum singular vector of matrix has measure zero.

---

### Official Review · Reviewer_asFh · 2023-11-06

**Soundness:** 2 fair
**Presentation:** 3 good
**Contribution:** 2 fair
**Rating:** 6
**Confidence:** 3

**Summary:**

This paper analyzes the linear stability of SGD through the lens of the loss surface geometry. Assuming a scenario where the model perfectly fits the data, the necessary and sufficient conditions for linear stability is characterized by learning rates, batch sizes, sharpness (the maximum eigenvalue of the Hessian), and a coherence measure of the individual loss Hessians, which is newly proposed in this work. The theoretical findings are validated through experiments on a synthetic optimization problem.

**Strengths:**

- The paper relaxes certain assumptions in characterizing the stability of SGD, e.g., the restriction to MSE loss, from existing works.
- The mathematical derivations appear to be sound and accurate.

**Weaknesses:**

- The paper lacks intuitive and qualitative explanations of the characterized linear stability, which could enhance its accessibility.
- The experiments are limited to engineered quadratic losses without considering actual neural networks or real-world data.
- The paper should address scenarios where the condition $\nabla_w l_i (w) = 0$ is violated. There may be various cases that $\| \nabla_w l_i \|$ is small but non-zero, e.g., cross-entropy loss without label smoothing, early stopping, and so on. How can the proposed analysis accommodate these cases?
- There are existing works to characterize the stability of SGD considering the noise covariance matrix $\Sigma = \frac{1}{n}\sum_i \nabla l_i(w) \nabla l_i(w)^T - \nabla L(w) \nabla L(w)^T$, without assuming $\nabla_w l_i (w) = 0$. The paper should clarify how its results relate to these existing works.

**Questions:**

- What is the exact notion of stability considered in the paper? It is not clearly explained in the manuscript.
- Is the coherence measure easily computable for typical neural networks? How complex would it be to compute in practice?
- On page 4, the term $x_i$ is used but not defined.
- It seems that the second last paragraph on page 4 assumes that $H_i$ is a matrix of rank one. Is this the case?

---

> ### Author Response · Authors · 2023-11-16
>
> We thank the reviewer for their insightful comments.
>
> > The paper lacks intuitive and qualitative explanations of the characterized linear stability, which could enhance its accessibility.
>
> We have made an effort to explain our intuitive understanding of our results within the page limits, and we note that Reviewer ELtf states: "Very well-written. I especially liked how the theoretical discussions are backed with intuitions and specific examples.". However, we understand that what is intuitive to us may not be to everyone. Hence, we could appreciate any more details on what specifically could be improved to make it accessible to a wider audience.
>
> > The experiments are limited to engineered quadratic losses without considering actual neural networks or real-world data.
>
> > Is the coherence measure easily computable for typical neural networks? How complex would it be to compute in practice?
>
> We note that our main contributions are theoretical and consider the general problem of minimizing additively decomposable loss functions. By restricting ourselves to these quadratic losses, we are able to avoid introducing unknown structure into the problem. This allows us to validate our general theory and get an idea of its tightness in the worst case.
>
> Computing the coherence measure naively would take about $O(n^2d^3)$ time, which is prohibitive for realistic neural networks. However, it likely can be approximated more efficiently using iterative methods, for example, the main bottleneck is computing $\||H_i^{1/2}H_j^{1/2}\||_F$ for all $i,j \in [n]$. However, we can estimate the Frobenius norm of a matrix product through matrix-vector products much more efficiently than it can be computed exactly.
>
> We do agree that extending this understanding to neural networks and real-world data is a critical next step, as we noted in our conclusion. However, we believe that handling these numerical challenges is best left as future work since our current paper has a different focus.
>
> > The paper should address scenarios where the condition $\nabla_w l_i (w) = 0$ is violated. There may be various cases that $| \nabla_w l_i |$ is small but non-zero, e.g., cross-entropy loss without label smoothing, early stopping, and so on. How can the proposed analysis accommodate these cases?
>
> We agree that the understanding the behavior of the case where the sample-wise gradients do not completely vanish is an important problem. At the same time, we would like to note that our assumption that $\nabla \ell_i(w^*) = 0$ for all $i \in [n]$ is a generally accepted assumption in this line of work (see e.g., [Ziyin at. al. 2023] and references within); and this is an important and practical regime to study.
>
> > There are existing works to characterize the stability of SGD considering the noise covariance matrix $\Sigma = \frac{1}{n}\sum_i \nabla l_i(w) \nabla l_i(w)^T - \nabla L(w) \nabla L(w)^T$, without assuming $\nabla_w l_i (w) = 0$. The paper should clarify how its results relate to these existing works.
>
> The work most relevant to ours that takes the approach of looking at the noise covariance matrix is [Wu et. al., 2022]. We have restated their main results in Appendix B and compare our results to theirs extensively in the related work section and in Section 3.2.1.
>
> Some important observations we repeat here are that [Wu et. al., 2022] is restricted to MSE loss, and for the conditions of their main theorem to hold, the optimal loss must be non-zero. This makes further comparison somewhat difficult, since their result does not apply to perfectly fit data as we consider and as is more common to consider in overparameterized networks.
>
> > What is the exact notion of stability considered in the paper? It is not clearly explained in the manuscript.
>
> In the paragraph before our contributions section, we have added the sentence: ``We consider mean-squared stability, that is, $w^*$ is considered unstable if iterates of SGD diverge from $w^*$ under the $\ell_2$-norm in expectation".
>
> > On page 4, the term $x_i$ is used but not defined.
>
> Thanks for pointing this out. We have fixed the expression.
>
> > It seems that the second last paragraph on page 4 assumes that is a matrix of rank one. Is this the case?
>
> Thank you for pointing out this omission. We have added this assumption in the example along with the following explanation for why it is reasonable.
>
> For binary classification with $f$ denoting a scalar scoring function of weights and loss being an $\mathbb{R} \rightarrow \mathbb{R}$ function of $f$, the Hessian is the sum of a rank-one term and a full second order term. However, in over-parameterized systems, each $f_i$ attains its least value and hence the full second order term is zero, and so $H_i$ being rank-one is a good assumption. (See eqn. (13) for additional details here: https://www.cs.toronto.edu/~rgrosse/courses/csc2541_2021/readings/L02_Taylor_approximations.pdf)

---

> > ### Comment · Reviewer_asFh · 2023-11-22
> >
> > Thanks for the detailed answers to the questions. I appreciate the theoretical contributions of the paper.
> >
> > After reading the authors' responses, I have additional comments:
> > - Even though the theorems concisely summarize the paper's main results, it would be better if some explanations existed of the intuitive mechanism of how incoherent Hessians (or small $\sigma$) can lead to diverging behavior. For example, can the overshoot of the second setting explained in Section 3.1 somehow lead to the diverging behavior?
> > - In [Wu et al., 2022]'s setting, even if $\nabla L(w)=0$ and $\nabla l_i(w)=0$, in their Lemma 2.2, the term $\frac{Tr(\Sigma(w)G(w))}{2L(w)\|G(w)\|_{F}^2}$ is lower-bounded by a term which can be positive. Hence, their analysis may also be valid when $\nabla l_i(w)=0$ case.
> > - In the additional sentence of the second last paragraph of Section 3.1 in the revised paper, 'binary cross-entropy' may be more appropriate than 'cross-entropy.'

---

> > > ### Author Response · Authors · 2023-11-22
> > > **Thank you for the additional comments!**
> > >
> > > We appreciate the additional valuable comments by the reviewer
> > >
> > > > Even though the theorems concisely summarize the paper's main results, it would be better if some explanations existed of the intuitive mechanism of how incoherent Hessians (or small $\sigma$) can lead to diverging behavior. For example, can the overshoot of the second setting explained in Section 3.1 somehow lead to the diverging behavior?
> > >
> > > Thanks for pointing us to where the confusion is. Note that the stability of GD is determined by whether \lambda_1(H) < 2/eta, where eta is the step size. We add the following sentence to clarify in the paragraph of our second setting in Section 3.1: "This overshoot causes instability, since it effectively increases the step size of SGD compared to GD in this scenario by a factor of n/B along the sampled directions."
> > >
> > > > In [Wu et al., 2022]'s setting, even if $\nabla L(w)=0$ and $\nabla l_i(w)=0$, in their Lemma 2.2, the term $\frac{Tr(\Sigma(w)G(w))}{2L(w)|G(w)|_{F}^2}$ is lower-bounded by a term which can be positive. Hence, their analysis may also be valid when $\nabla l_i(w)=0$ case.
> > >
> > > We note that the complexity measure $\mu$ is undefined when $L(w) = 0$. Hence, the conditions of Theorem 1 cannot be satisfied when $L(w^*) = 0$. We do agree that it is likely that this singularity could possibly be avoided. However, we did not see an immediate way to modify the proof to fix this issue, and so the theoretical statment in [Wu et al., 2022] cannot be compared in our setting as written.
> > >
> > > > In the additional sentence of the second last paragraph of Section 3.1 in the revised paper, 'binary cross-entropy' may be more appropriate than 'cross-entropy.'
> > >
> > > Thanks for the recommendation. We will make the change.

---

> > > > ### Comment · Reviewer_asFh · 2023-11-22
> > > >
> > > > Thank you for the additional clarifications.
> > > >
> > > > I still struggle to intuitively understand how the SGD becomes unstable in the incoherent Hessian setting.
> > > > Can you elaborate on why the effective increase in stepsize along the sampled directions can lead to instability?
> > > > In the second setting of Section 3.1, over multiple updates, the effective stepsize of the SGD along each sample direction will be the same as that of GD on average. Is the consecutive overshooting along different sample directions occurring in each update the problem?

---

> > > > > ### Author Response · Authors · 2023-11-23
> > > > >
> > > > > We appreciate the opportunity to clarify any miscommunication on the intuition of our results. The quick answer is that (in setting 2) when the $i$-th Hessian is sampled, we overshoot on the $i$-th coordinate by a factor of $\frac{n}{B}$. When the $i$-th coordinate is not sampled, it does not change at all. Below we give a more detailed explanation of this situation.
> > > > >
> > > > > Let $\\{H\_i\\}\_{i \in [n]}$ be such that $H_i$ equals $n$ in the $i$-th diagonal entry and zero everywhere else. Let $\ell_i(w) = w^TH_iw$ be the "sample-wise" loss functions and $L(w) = \frac{1}{n} \sum_{i=1}^n \ell_i(w)$ be the overall loss function. If we let $w_t$ be the iterate under regular gradient descent, then the first coordinate would diverge if $\eta > 2$. Note that $[\nabla L(w)]_i = 1$ for all $i \in [n]$.
> > > > >
> > > > > Now let us consider SGD with respect to the first coordinate of $w_t$. First, suppose the first coordinate is not sampled, then the stochastic loss $\frac{1}{B}\sum_{i\in\mathcal{S}} \ell\_i(w)$ does not depend on the first entry of $w$, and so the first coordinate doesn't change. If the first coordinate is sampled, i.e., $1 \in \mathcal{S}$, then $\frac{\partial}{\partial [w]\_1}\frac{1}{B}\sum\_{i\in\mathcal{S}} \ell_i(w) = \frac{1}{B} \frac{\partial}{\partial [w]_1} \ell_1(w) = \frac{n}{B}$. So the first coordinate will change by $\frac{n}{B} \cdot \eta$.
> > > > >
> > > > > Intuitively, one can see that if $i \in \mathcal{S}$, then the change to the $i$-th coordinate will be $\frac{n}{B}$ times what it would be under GD. If $i \not \in \mathcal{S}$, then the $i$-th coordinate will not change at all. Therefore, the convergence/divergence behavior of the $i$-th coordinate is identical to that of GD with its learning multiplied by an $\frac{n}{B}$ factor. The overall divergence of $w_t$ just needs one coordinate to diverge.

---

> > > > > > ### Comment · Reviewer_asFh · 2023-11-23
> > > > > >
> > > > > > Thank you for the clarification. I could understand the idea further. I will increase my rating.

---

### Author Response · Authors · 2023-11-17
**Revision uploaded**

We have uploaded a revision of the paper containing the changes we described in our responses.

---

### Author Response · Authors · 2023-11-22
**Reminder to address our comments**

Dear Reviewers,

Thanks for your detailed reviews. We are yet to receive comments on our replies to your initial reviews. Gentle reminder to respond, we would be happy to address any other feedback.

Thanks,

Authors

---

### Meta-Review · Area_Chair_UKMH · 2024-01-06

**Metareview:**

The work investigates stability of SGD iterates under linearized dynamics, a topic that has been investigated in the recent literature. The analysis is based on sharpness and, more generally, properties of the Hessian of the loss. The work presents conditions, based on a new Hessian coherence measure and related technical results, on when iterates will diverge or converge. The reviewers mostly appreciate the technical advances in the paper, including the improvements made over recent related work on the theme, e.g., extensions beyond square loss, arguably more relaxed conditions for the analysis, etc. The reviewers also share some concerns, including the difficulty of intuitively understanding implications of the results, discrepancies between SGD sampling used in theory vs. experiments, need for sample gradients to vanish, etc. The authors have given detailed responses to many of these concerns, which have been acknowledged by the reviewers. The manuscript will be stronger by suitably including some of the discussions in the main paper, e.g., as remarks.

**Justification For Why Not Higher Score:**

The support from some of the reviewers were a bit luke-warm as some of the concerns possibly did not get fully resolved to their satisfaction.

**Justification For Why Not Lower Score:**

The work arguably makes non-trivial progress on a theoretical theme of current interest.

---

### Decision · Program_Chairs · 2024-01-16

Accept (poster)